# Near-Optimal Regret for Policy Optimization in Contextual MDPs with General Offline Function Approximation

**Orin Levy** [1]  **Aviv Rosenberg** [2]  **Alon Cohen** [2 3]  **Yishay Mansour** [1 2]

## Abstract

We introduce `OPO-CMDP`, the first policy optimization algorithm for stochastic Contextual Markov Decision Process (CMDPs) under general offline function approximation. Our approach achieves a high probability regret bound of $\widetilde{O}(H^4\sqrt{T|S||A|\log(|\mathcal{F}||\mathcal{P}|)})$, where $S$ and $A$ denote the state and action spaces, $H$ the horizon length, $T$ the number of episodes, and $\mathcal{F}, \mathcal{P}$ the finite function classes used to approximate the losses and dynamics, respectively. This is the first regret bound with optimal dependence on $|S|$ and $|A|$, directly improving the current state-of-the-art (Qian, Hu, and Simchi-Levi, 2024). These results demonstrate that optimistic policy optimization provides a natural, computationally superior and theoretically near-optimal path for solving CMDPs.

## 1. Introduction

*Contextual Markov Decision Processes* (CMDPs) (Hallak et al., 2015) extend the classical *Reinforcement Learning* (RL) framework of *Markov Decision Processes* (MDPs) to incorporate external factors, called *context*, that influence the agent–environment interaction in each episode. Such an extension is essential in many modern online decision-making problems, where the environment varies with observed side information, such as user attributes in recommendation systems, patient profiles in healthcare, or task specifications in personalized robotics and dialog systems.

In the CMDP model, the interaction proceeds over $T$ episodes. At the beginning of each episode, the learner observes a freshly sampled context $c$ and selects a behavioral strategy, or *policy*, $\pi$, and then interacts with the environment for $H$ steps. The performance of the learner is evaluated through the notion of *regret*, defined as the difference between the expected cumulative loss incurred by the agent over $T$ episodes and the expected cumulative loss achieved by the optimal context-dependent policy.

When the context space is large, CMDPs exhibit an extreme exploration-exploitation trade-off: in each episode, the learner may encounter a context that induces a substantially different MDP, rendering trajectories collected under previous contexts only weakly informative. For *stochastically drawn* contexts, a natural approach is to encode the context into the state space and adjust the transition dynamics accordingly; however, this yields a joint context-state space whose size scales with the cardinality of the context space, leading to regret bounds that depend explicitly on it. Similarly, learning an independent MDP for each context is infeasible, as it incurs regret that grows linearly with the number of contexts. These limitations emphasize the need to generalize across contexts: to achieve sublinear regret without direct dependence on the context space cardinality, the learner must exploit shared structure to infer the dynamics of an unseen context from trajectories collected under previously observed contexts.

To obtain this generalization capability, the CMDP literature (Levy and Mansour, 2022; 2023; Levy et al., 2024; Qian et al., 2024) adopts the *offline function approximation* framework, which assumes that both the loss functions and the transition dynamics admit realizable representations within known function classes that are accessed via a regression oracle. This framework enables generalization across contexts while preserving computational tractability.

Offline function approximation is not only a theoretical necessity for CMDPs but also the foundation of applicable modern large-scale RL solutions that train highly expressive neural networks. *Policy Optimization* (PO) methods form one of the most practical and empirically successful families of RL algorithms, and specifically popular algorithms like TRPO (Schulman et al., 2016), PPO (Schulman et al., 2017) and SAC (Haarnoja et al., 2018). Such methods have achieved major advances across multiple domains includ-

---

[1]School of Computer Science and AI, Tel Aviv University [2]Google Research [3]School of Electrical Engineering, Tel Aviv University. Correspondence to: Orin Levy <orinlevy@mail.tau.ac.il>, Aviv Rosenberg <avivros007@gmail.com>, Alon Cohen <alonco@tauex.tau.ac.il>, Yishay Mansour <mansour.yishay@gmail.com>.

*Proceedings of the 43$^{rd}$ International Conference on Machine Learning*, Seoul, South Korea. PMLR 306, 2026. Copyright 2026 by the author(s).

*Table 1.* Comparison of regret bounds for stochastic CMDPs with offline function approximation. We note that (Qian et al., 2024) assume time-variant dynamics (i.e., the dynamics depend on the horizon) and thus the CMDP is layered but not loop free. In addition, they use a single function class $\mathcal{M}$ to approximate both the dynamics and the reward, such that $|\mathcal{M}| = |\mathcal{F}||\mathcal{P}|$.

| Algorithm | Structural Assumptions | Loop-Free? | Regret Bound |
|---|---|---|---|
| RM-UCDD (Levy and Mansour, 2023) | Minimum reachability $p_{\min}$ | ✓ | $\widetilde{O}\left(\sqrt{p_{\min}^{-2}H^2|S|^3|A|\,T\log(|\mathcal{F}||\mathcal{P}|/\delta)}\right)$ |
| E-UC$^3$RL (Levy et al., 2024) | Eluder dimension $d$ | ✓ | $\widetilde{O}\left(\sqrt{dH^6|S||A|\,T\log(|\mathcal{F}||\mathcal{P}|/\delta)}\right)$ |
| LOLIPOP (Qian et al., 2024) | None | ✗ | $\widetilde{O}\left(\sqrt{H^7|S|^4|A|^3\,T\log(|\mathcal{F}||\mathcal{P}|/\delta)}\right)$ |
| **OPO-CMDP (this work)** | None | ✓ | $\widetilde{O}\left(\sqrt{H^8|S||A|\,T\log(|\mathcal{F}||\mathcal{P}|/\delta)}\right)$ |

ing robotics (Peters and Schaal, 2006; 2008; Levine et al., 2016; Gu et al., 2017), continuous control (Mnih et al., 2015; Schulman et al., 2016; Lillicrap et al., 2016), and large-scale human-in-the-loop systems, where policy optimization underlies Reinforcement Learning from Human Feedback (RLHF) for fine-tuning Large Language Models (Stiennon et al., 2020; Glaese et al., 2022; Ouyang et al., 2022). In addition to their empirical success, PO algorithms provide closed-form stochastic policy updates that naturally balance exploration and exploitation, often through exponential weighting update rules, and have become the dominant choice in large-scale systems due to their stability and ease of implementation. However, despite their popularity in practice and good algorithmic properties, PO methods remain theoretically underexplored in the context of CMDPs with general offline function approximation.

Current state-of-the-art in this setting (LOLIPOP; Qian et al., 2024) achieves a rate-optimal regret bound of $\widetilde{O}(\sqrt{T})$. However, they use a non-practical algorithm and end up with a high polynomial dependency in the number of states and actions. Our work addresses both of these issues, introducing a PO-based algorithm with strong theoretical guarantees.

**Main contributions.** We propose OPO-CMDP, the first policy optimization algorithm for stochastic CMDPs under general offline function approximation, relying only on standard offline regression oracles. We prove that OPO-CMDP achieves $\widetilde{O}(H^4\sqrt{T|S||A|\log(|\mathcal{F}||\mathcal{P}|)})$ regret, where $T$ is the number of episodes, $H$ is the horizon length, $S$ and $A$ are the state and action spaces, respectivly and $\mathcal{F}, \mathcal{P}$ are finite and realizable function classes use to estimate the loss and dynamics, respectively. Our results improve the dependence on $|S|$ and $|A|$ over prior work (Qian et al., 2024) and match the known lower bound (Levy and Mansour, 2023) in all the parameters, except for $H$. Our analysis develops new confidence bounds for stochastic policies that remove the need for additional structural assumptions such as finite Eluder dimension (Levy et al., 2024) or minimum reachability (Levy and Mansour, 2023). Our confidence bound analysis, which leverages function-approximation-based estimators for losses and dynamics, may be of independent interest.

### 1.1. Related Literature

**Contextual RL.** Early work in CMDPs focused on specific structural assumptions, including small hidden context spaces (Hallak et al., 2015), low Bellman-rank (Jiang et al., 2017), Witness Rank (Sun et al., 2019), and linear/GLM mixtures (Modi et al., 2018; Modi and Tewari, 2020; Deng et al., 2024). Recent literature has shifted toward general function approximation, categorized by the available regression oracle. While adversarial CMDPs rely on online oracles (Foster et al., 2021; Levy et al., 2023), our work focuses on the stochastic setting with offline regression oracles.

Following the initial reduction to ERM by Levy and Mansour (2022), subsequent works achieved $\widetilde{O}(\sqrt{T})$ regret under additional assumptions like minimum reachability (Levy and Mansour, 2023) or finite Eluder dimension (Levy et al., 2024). In addition, Levy and Mansour (2023) proved a lower bound of $\widetilde{\Omega}(\sqrt{HT|S||A|\log|\mathcal{G}|})$ for CMDPs with known dynamics, where $\mathcal{G}$ is a function class used to approximate the contextual rewards.

**Comparison with current state-of-the-art.** The current state-of-the-art algorithm, LOLIPOP (Qian et al., 2024), studies layered CMDPs with time-varying transition dynamics $P = \{P_h\}_{h=1}^H$. In their setting, the state space is shared across layers while transition probabilities differ between layers, which allows loops to occur. In contrast, we consider a layered and loop-free structure. As a consequence, the resulting regret bounds are not directly comparable in their dependence on the horizon $H$.

Algorithmically, LOLIPOP relies on offline density estimation over a class of CMDPs, whereas our approach only requires standard offline regression oracles for dynamics and loss estimation. Moreover, LOLIPOP employs a layer-by-layer exploration strategy, maintains an explicit cover over policies and set of trusted occupancy measures. In each episode, they first sample a policy from this cover using inverse gap weighting distribution that is computed using the trusted occupancy measures, and then executing the sampled policy to generate new trajectory. This action-selection mechanism, together with the maintenance

and refinement of the policy cover, introduces substantial computational and conceptual complexity; Each refinement stage in `LOLIPOP` requires solving offline density estimation subproblems to construct trusted occupancy measures, resulting in a multi-layer, epoch-based procedure with significant overhead. This complexity is also reflected in regret bounds with high polynomial dependence on $|S|$ and $|A|$, stemming from the need to explore and control uncertainty separately across layers.

In contrast, our approach is based on explicit optimistic policy optimization update rule. For each observed context, actions are sampled directly from a single evolving policy, and exploration is incorporated naturally through the policy update itself and an additional exploration bonus. As a result, our algorithm admits a significantly simpler implementation, lower per-iteration computational cost, and a more transparent exploration-exploitation mechanism. Furthermore, our approach improves the dependence on $|S|$ and $|A|$ while achieving comparable statistical guarantees.

**Contextual Bandits.** Contextual Multi-Armed Bandits (CMABs) represent a single-state instance of CMDPs. Research has evolved from inefficient finite-policy algorithms (Auer et al., 2002; Agarwal et al., 2014) to efficient methods using offline regression oracles (Agarwal et al., 2012; Foster et al., 2018; Simchi-Levi and Xu, 2022). Our work extends the policy optimization approach recently proposed for CMABs (Levy and Mansour, 2025) to the full MDP setting.

**Policy optimization in RL.** The theoretical foundations of policy optimization are well-established for tabular MDPs. Early work by Even-Dar et al. (2009) introduced weighted-majority methods for known dynamics, which were later adapted to the bandit-feedback setup by Neu et al. (2010). For the more challenging setting of unknown dynamics, Shani et al. (2020) proposed an optimistic PO framework, while Luo et al. (2021) achieved rate-optimality in fully adversarial environments by incorporating sophisticated exploration bonuses. Recent advances have further extended these techniques to handle practical complexities such as delayed (Lancewicki et al., 2022; 2023) or aggregated feedback (Cassel et al., 2024; Lancewicki and Mansour, 2025). Beyond the tabular case, PO has been successfully applied to linear MDPs, with the current state-of-the-art attaining rate-optimal $\widetilde{O}(\sqrt{T})$ regret (Sherman et al., 2024; Cassel and Rosenberg, 2024). Despite these developments, the application of pure policy optimization has remained largely restricted to tabular and linear structures. Our work bridges this gap, providing the first extension of the policy optimization framework to general offline function approximation within the Contextual MDP setup.

## 2. Preliminaries

### 2.1. Episodic Loop-Free MDP

An MDP is defined by a tuple $(S, A, P, \ell, s_1, H)$, where $S$ and $A$ are finite state and action sets, $s_1 \in S$ is the initial state, $H$ is the horizon, $\ell : S \times A \to [0, 1]$ is the loss function and $P : S \times A \to \Delta_S$ is the transition function.

An episode consists of a sequence of $H$ interactions. At step $h$, if the environment is at state $s_h$ and the agent performs action $a_h$, then the environment transitions to state $s_{h+1} \sim P(\cdot|s_h, a_h)$, and the agent receives a loss $L(s_h, a_h) \in [0, 1]$, sampled independently from a fixed distribution $\mathcal{D}_{s_h, a_h}$ that satisfies $\ell(s_h, a_h) = \mathbb{E}_{\mathcal{D}_{s_h, a_h}}[L(s_h, a_h)]$.

For technical convenience and without loss of generality, we assume that the state space and accompanying transition probabilities have a layered, loop-free structure. Concretely, we assume that the state space can be decomposed into $H + 1$ disjoint subsets (layers) $S_1, \ldots, S_H, S_{H+1}$, such that transitions are only possible between consecutive layers; that is, for $h' \neq h + 1$ we have $P(s_{h'} \mid s_h, a) = 0$ for all $s_{h'} \in S_{h'}, s_h \in S_h, a \in A$. In addition, $S_1 = \{s_1\}$ and $S_{H+1} = \{s_{H+1}\}$, meaning there are unique start and end states, with the final loss at the end state equal to $0$.

A *stationary policy* $\pi : S \to \Delta_A$ is a mapping that gives the probability $\pi(a \mid s)$ to take action $a$ when visiting state $s$. The *value function* of $\pi$ in MDP $M$ represents its expected cumulative loss $V_{M,h}^\pi(s) = \mathbb{E}_{\pi,P}[\sum_{k=h}^H \ell(s_k, a_k)|s_h = s]$. Similarly, the state-action *Q-function* is defined as $Q_{M,h}^\pi(s, a) = \mathbb{E}_{\pi,P}[\sum_{k=h}^H \ell(s_k, a_k) \mid s_h = s, a_h = a]$. For brevity, we define $V_{M,H+1}^\pi(s) = 0$, $Q_{M,H+1}^\pi(s, a) = 0$ and denote $V_M^\pi(s_1) := V_{M,1}^\pi(s_1)$ (we omit the $M$ subscript when clear from context). The value and Q-function are related via the Bellman equations:

$$Q_{M,h}^\pi(s, a) = \ell(s, a) + PV_{M,h+1}^\pi(s, a);$$
$$V_{M,h}^\pi(s) = \langle Q_{M,h}^\pi(s, \cdot), \pi(\cdot \mid s) \rangle, \tag{1}$$

where $\langle \cdot, \cdot \rangle$ is the inner product and $PV_h(s, a) = \mathbb{E}_{s' \sim P(\cdot|s,a)}[V_h(s')]$. Finally, throughout the paper we make use of *occupancy measures* (Puterman, 2014; Zimin and Neu, 2013): let $q_h(s, a \mid \pi, P)$ be the probability of reaching state $s$ and performing action $a$ at time $h$ in an episode generated using policy $\pi$ and dynamics $P$. Conveniently, the occupancy measure can be computed efficiently (given $\pi$ and $P$), and it satisfies $V_M^\pi(s_1) = \langle q(\cdot, \cdot \mid \pi, P), \ell \rangle$.

### 2.2. Stochastic Contextual MDP (CMDP)

A *Contextual Markov Decision Process (CMDP)* is defined by a tuple $(\mathcal{C}, S, A, \mathcal{M})$, where $\mathcal{C}$ is the context space, and $S, A$ the state and action spaces. $\mathcal{M}$ maps a context $c \in \mathcal{C}$ to an episodic loop-free MDP $\mathcal{M}(c) = (S, A, P_c^\star, \ell_c^\star, s_1, H)$. We assume without loss of generality that all contexts share

the same layered structure, while the transition probabilities may differ across contexts.

We consider *stochastic* CMDPs, meaning that contexts are sampled i.i.d. from an unknown fixed distribution $\mathcal{D}$. For mathematical convenience, we assume that the context space $\mathcal{C}$ is finite but potentially huge. Our bounds do not depend on the size of the context space and can be further extended to the infinite case, using appropriate dimension or discretization (see e.g., Shalev-Shwartz and Ben-David, 2014).

A *context-dependent policy* $\pi : \mathcal{C} \times S \to \Delta_A$ gives the probability $\pi_c(a \mid s)$ to take action $a$ in state $s$ under context $c$. The interaction between the agent and the environment is defined as follows. In the beginning of each episode $t = 1, 2, \ldots, T$ the agent observes a context $c_t \sim \mathcal{D}$ and chooses a context-dependent policy $\pi^t$. The policy $\pi^t_{c_t}$ is then executed in the MDP $\mathcal{M}(c_t)$ and the agent observes a trajectory $\sigma_t := (c_t, s^t_1, a^t_1, \ell^t_1, \ldots, s^t_{H+1})$. Our goal is to minimize the (pseudo) *regret*, defined as

$$\mathcal{R}_T := \sum_{t=1}^{T} V^{\pi^t_{c_t}}_{c_t}(s_1) - V^{\pi^\star_{c_t}}_{c_t}(s_1),$$

where $V^{\pi_c}_c(s_1) = V^{\pi_c}_{\mathcal{M}(c),1}(s_1)$ and $\pi^\star$ is an optimal context-dependent policy, satisfying for all $c \in \mathcal{C}$ that $\pi^\star_c \in \arg\min_{\pi:S\to\Delta_A}\{V^\pi_c(s_1)\}$ (See, e.g., Sutton and Barto, 2018; Mannor et al., 2022). In addition, we consider the *expected (pseudo) regret*, where the expectation is over the contexts, i.e.,

$$\mathbb{E}\mathcal{R}_T := \sum_{t=1}^{T} \mathbb{E}_{c_t \sim \mathcal{D}}\left[ V^{\pi^t_{c_t}}_{c_t}(s_1) - V^{\pi^\star_{c_t}}_{c_t}(s_1) \right].$$

### 2.3. Offline Function Approximation

In order to derive regret bounds that are independent of the context space size $|\mathcal{C}|$, we use function approximation assumptions that are standard in the CMDP literature (Levy and Mansour, 2023; Levy et al., 2024; Qian et al., 2024). Our first assumption is *realizability*, i.e., we assume access to finite function classes $\mathcal{F} \subseteq \mathcal{C} \times S \times A \to [0, 1]$ and $\mathcal{P} \subseteq \mathcal{C} \times S \times A \to \Delta_S$ such that there exists $f^\star \in \mathcal{F}$ satisfying $f^\star_c(s, a) = \ell^\star_c(s, a)$ for all $(c, s, a) \in \mathcal{C} \times S \times A$, and $P^\star \in \mathcal{P}$. For simplicity, we assume that every dynamics in this class respects the true layered transitions structure.

Our second assumption is access to *offline regression oracles*. Given a dataset of trajectories $D = \{\sigma^i\}^t_{i=1}$, we assume access to offline oracles that solve Least Squares Regression and Log Loss Regression, respectively:

$$\hat{f}^{t+1} \in \arg\min_{f\in\mathcal{F}} \sum_{i=1}^{t}\sum_{h=1}^{H} \left( f_{c_i}(s^i_h, a^i_h) - \ell^i_h \right)^2.$$

$$\hat{P}^{t+1} \in \arg\max_{P\in\mathcal{P}} \sum_{i=1}^{t}\sum_{h=1}^{H} \log P_{c_i}(s^i_{h+1} \mid s^i_h, a^i_h).$$

Assume w.l.o.g that ties are broken deterministically. While the oracles can always iterate over the function class, some function classes admit efficient solutions (e.g., linear functions). Moreover, these oracles provide generalization guarantees which we will later use to bound the regret. For loss estimation, the least squares oracle gives the following guarantee.

**Corollary 2.1** (Corollary 4.2 in Levy et al., 2024). *Let $\hat{f}^t \in \mathcal{F}$ be the least squares minimizer. For any $\delta \in (0, 1)$, it holds with probability at least $1 - \delta$,*

$$\mathbb{E}_c\left[ \sum_{i=1}^{t-1} \mathcal{E}^t_{sq}(\pi^i, c) \right] \le 68H\log(2T^3|\mathcal{F}|/\delta), \qquad \forall t \ge 1,$$

*where $\mathcal{E}^t_{sq}(\pi, c)$ is the expected squared error at round $t$,*

$$\mathcal{E}^t_{sq}(\pi, c) = \mathbb{E}_{\pi_c, P^\star_c}\left[ \sum_{h=1}^{H} \left( \hat{f}^t_c(s_h, a_h) - f^\star_c(s_h, a_h) \right)^2 \;\Big|\; s_1 \right].$$

For dynamics estimation, the log loss oracle provides generalization guarantees with respect to the squared Hellinger distance (Foster et al., 2021).

**Definition 2.2** (Squared Hellinger Distance). For any two distributions $\mathbb{P}, \mathbb{Q}$ over a discrete support $X$, the Squared Hellinger Distance is given by

$$D^2_H(\mathbb{P}, \mathbb{Q}) = \sum_{x\in X} \left( \sqrt{\mathbb{P}(x)} - \sqrt{\mathbb{Q}(x)} \right)^2.$$

**Corollary 2.3** (Corollary 4.3 in Levy et al., 2024). *Let $\hat{P}^t \in \mathcal{P}$ be the log loss minimizer. For any $\delta \in (0, 1)$, it holds with probability at least $1 - \delta$,*

$$\mathbb{E}_c\left[ \sum_{i=1}^{t-1} \mathcal{E}^t_H(\pi^i, c) \right] \le 2H\log(TH|\mathcal{P}|/\delta), \qquad \forall t \ge 1,$$

*where $\mathcal{E}^t_H(\pi, c)$ is the expected squared Hellinger error at round $t$:*

$$\mathcal{E}^t_H(\pi, c) = \mathbb{E}_{\pi_c, P^\star_c}\left[ \sum_{h=1}^{H} D^2_H(P^\star_c(\cdot|s_h, a_h), \hat{P}^t_c(\cdot|s_h, a_h)) \;\Big|\; s_1 \right].$$

## 3. Main Result: Provable Policy Optimization Algorithm for CMDPs

Our algorithm extends the optimistic policy optimization (OPO) framework of Shani et al. (2020) to CMDPs by incorporating optimistic, function-approximation-based estimators for both the transition dynamics and the loss functions. The core idea of OPO is to update the policy via an exponential weights update using the most up-to-date estimate of an optimistic $Q$-function. This optimistic approximation

is constructed using tabular estimates of the transition dynamics and losses, augmented with exploration bonuses that account for approximation errors in both components. The optimistic $Q$-function is then computed using an optimistic loss estimate, which subtracts the corresponding exploration bonuses from the empirical mean loss.

We extend this approach to CMDPs. Our key insight lies in the design of the exploration bonus. Inspired by prior work on optimistic exploration in CMDPs with general function approximation (Xu and Zeevi, 2020; Levy and Mansour, 2023; Levy et al., 2024), which identified a fundamental property of contextual occupancy measures, we observe that for any context $c \in \mathcal{C}$, round $t \in [T]$, layer $h \in [H]$, and state-action pair $(s, a) \in \mathcal{S}_h \times \mathcal{A}$, the quantity $\sum_{i=1}^{t-1} q_h(s, a \mid \pi_c^i, P_c^\star)$ equals the expected number of visits to $(s, a)$ under context $c$ up to round $t$. Consequently, this quantity serves as a measure of the quality of the estimates $\hat{P}_c^t$ and $\hat{f}_c^t$ for that context at round $t$. A natural choice of exploration bonus is therefore inversely proportional to this occupancy measure. However, since the true transition dynamics $P_c^\star$ are unknown, this quantity cannot be computed exactly and must itself be approximated.

In Levy and Mansour (2023), this challenge is addressed by imposing a minimum reachability assumption, which allows the occupancy term to be replaced by the policy's action-selection probability. Subsequently, Levy et al. (2024) extend this approach by introducing Eluder-based counterfactual bonuses. At round $t$, they consider the cumulative occupancy of past policies evaluated under their contemporaneous learned dynamics, $\sum_{i=1}^{t-1} q_h(s, a \mid \pi_c^i, \hat{P}_c^i)$, and propose exploration bonuses inversely proportional to this quantity. Since this occupancy depends directly on potentially unstable offline regression estimates of the dynamics, they leverage the Eluder dimension assumption to stabilize the sequence of predicted transition models.

Inspecting the bonuses proposed in prior work, it is tempting to evaluate the occupancy measures of past policies under the most recent dynamics estimate $\hat{P}_c^t$, as it provides the most accurate approximation of the true dynamics. This intuition motivates our bonus design. We show that it suffices to construct exploration bonuses proportional to $\sum_{i=1}^{t-1} q_h(s, a \mid \pi_c^i, \hat{P}_c^t)$, where the most up-to-date dynamics estimate $\hat{P}_c^t$ is used uniformly to evaluate all policies required to compute the $t$'th policy, $\pi^t$. Formally, for a fixed context $c$, round $t$, layer $h$, and state-action pair $(s, a) \in \mathcal{S}_h \times \mathcal{A}$, we define the exploration bonus as

$$b_\beta^t(c, s, a) = \min\left\{1, \frac{\beta/2}{1 + \sum_{i=1}^{t-1} q_h(s, a \mid \pi_c^i, \hat{P}_c^t)}\right\}, \quad (2)$$

where $\beta \in \{\beta_\ell, \beta_P\}$ is an exploration parameter corresponding to the loss and dynamics function classes. We choose $\beta_\ell, \beta_P = O(\sqrt{T})$, which yields bonuses of order

---

**Algorithm 1** OPO-CMDP

1: **Input:** learning rate $\eta$, number of episodes $T$, tuning parameters $\beta_\ell, \beta_P$.
2: **Initialization:** $\pi^1$ is the uniform distribution for all $(c, s) \in \mathcal{C} \times S$. $\hat{f}_1, \hat{P}_1$ are arbitrary functions returned by the oracles on the empty dataset.
3: Observe the first context $c_1 \sim \mathcal{D}$.
4: **for** $t = 1, \ldots, T$ **do**
5:     Play policy $\pi_{c_t}^t$ and observe trajectory $\sigma_t$.
6:     Observe the next context $c_{t+1} \sim \mathcal{D}$.
7:     **Model Approximation (using offline oracles):**

$$\hat{f}^{t+1} \in \arg\min_{f \in \mathcal{F}} \sum_{i=1}^{t} \sum_{h=1}^{H} (f_{c_i}(s_h^i, a_h^i) - \ell_h^i)^2.$$

$$\hat{P}^{t+1} \in \arg\max_{P \in \mathcal{P}} \sum_{i=1}^{t} \sum_{h=1}^{H} \log P_{c_i}(s_{h+1}^i | s_h^i, a_h^i).$$

8:     **for** $k = 1, 2, \ldots, t$ **do**
9:       **Bonus Calculation:**
      For all $(s, a) \in S \times A$ compute

$$b_{c_{t+1}}^k(s, a) = b_{\beta_\ell}^k(c_{t+1}, s, a) + 2H b_{\beta_P}^k(c_{t+1}, s, a)$$
$$\hat{\ell}_{c_{t+1}}^k(s, a) = \hat{f}_{c_{t+1}}^k(s, a) - b_{c_{t+1}}^k(s, a),$$

      where $b_\beta^k(c, s, a)$ is defined in Equation (2).
10:       Set $\hat{V}_{c_{t+1}, H+1}^k(s_{H+1}) = 0$.
11:       **for** $h = H, H-1, \ldots, 1$ **do**
12:         **Policy Evaluation:**
        For all $(s, a) \in S_h \times A$ compute

$$\hat{Q}_{c_{t+1}, h}^k(s, a) = \max\Big\{0,$$
$$\hat{\ell}_{c_{t+1}}^k(s, a) + \hat{P}_{c_{t+1}}^k \hat{V}_{c_{t+1}, h+1}^k(s, a)\Big\}$$
$$\hat{V}_{c_t, h}^k(s) = \langle \pi_{c_t}^k(\cdot|s), \hat{Q}_{c_t, h}^k(s, \cdot)\rangle.$$

13:       **end for**
14:       **Policy Improvement:**
      For all $(s, a) \in S \times A$ compute

$$\pi_{c_{t+1}}^{k+1}(a|s) \propto \pi_{c_{t+1}}^k(a|s) \exp\left(-\eta \hat{Q}_{c_{t+1}, h}^k(s, a)\right).$$

15:     **end for**
16: **end for**

---

$O(1/\sqrt{T})$, matching the exploration rates known for tabular MDPs (Jaksch et al., 2010).

As our bonuses depend on past context-dependent policies, they require evaluating $\pi_{c_t}^k$ for all $k \in [t-1]$ at round $t$. Policy optimization is particularly well suited for this purpose, as it admits closed-form updates and avoids solving

additional optimization problems for each computation (in contrast to Levy and Mansour, 2023; Levy et al., 2024).

Finally, we describe our algorithm, which is fully specified in Algorithm 1. The algorithm initializes with a uniform policy over all contexts and states. At each round $t \geq 2$, we first estimate the loss and transition dynamics by applying least-squares and log-loss regression oracles, respectively, to all previously observed trajectories. We denote the resulting loss estimator by $\hat{f}^t$ and dynamics estimator by $\hat{P}^t$.

Next, upon observing a fresh context $c_t$, we iteratively apply policy optimization updates to compute all past policies of $c_t$. These policies are used both to construct the exploration bonuses for the subsequent update and to update the current policy via an exponential update step.

Importantly, policies are not precomputed for all contexts; instead, they are generated on demand only for the contexts encountered during interaction. Hence, the algorithm reduces to standard optimistic policy optimization when conditioned on a fixed context, yielding a natural and computationally efficient extension of the framework to CMDPs with general function approximation.

Our main result is a regret analysis of this algorithm, which is presented in the following theorem. Our regret bound is optimal in all parameters except for the horizon $H$, matching the known lower bound for CMDPs with function approximation (Levy and Mansour, 2023) up to a factor of $H^{3.5}$. Moreover, our result improves the dependence on $|S|$ and $|A|$ compared to the previous state-of-the-art (Qian et al., 2024).

**Theorem 3.1.** *Suppose that we run* OPO-CMDP *(Algorithm 1) with the parameters defined in Theorem A.12 (Appendix A.3). Then, with probability at least $1 - \delta$, we have*

$$\mathcal{R}_T \leq O\Big(H^4 \sqrt{T|S||A| \log(T+1) \log(TH|\mathcal{F}||\mathcal{P}|/\delta)}\Big).$$

## 4. Regret Analysis

In this section, we present the regret analysis of Algorithm 1, establishing Theorem 3.1. Since a high-probability regret bound follows directly from a high-probability bound on the expected regret (see Corollary B.7), we focus our efforts on bounding the expected regret. Throughout the analysis, we state bounds informally using the notation $\lesssim$, which suppresses numeric constants, logarithmic factors, and lower-order terms for clarity. Full formal statements can be found in Appendix A.

### 4.1. Proof Outline

Below, we outline the decomposition of the expected regret, highlighting the main components of our analysis.

Using the standard regret decomposition, we have the following:

$$\mathbb{ER}_T = \sum_{t=1}^{T} \mathbb{E}_{c_t}\left[ V_{c_t}^{\pi_{c_t}^t}(s_1) - \hat{V}_{c_t}^t(s_1) \right] \quad (i)$$

$$+ \sum_{t=1}^{T} \mathbb{E}_{c_t}\left[ \hat{V}_{c_t}^t(s_1) - V_{c_t}^{\pi_{c_t}^\star}(s_1) \right] \quad (ii),$$

where $(i)$ stands for the approximation error of the CMDP, and $(ii)$ is the optimistic term which will require more steps later. We bound each term separately, starting from $(i)$.

**Lemma 4.1.** *With probability at least $1 - \delta/4$,*

$$(i) \lesssim \left( \frac{1}{\beta_\ell} + \frac{H}{\beta_P} \right) TH^3 \log(TH|\mathcal{F}||\mathcal{P}|/\delta)$$

$$+ \mathbb{E}_c\left[ \sum_{t=2}^{T} \sum_{h,s,a} q_h(s,a|\pi_c^t, P_c^\star) b_c^t(s,a) \right].$$

The above lemma is mainly derived by the oracle's concentration guarantees (see complete statement and proof in Lemma A.1). Importantly, it bounds the approximation error using the expected sum of our exploration bonuses with respect to the executed policies. Our novel technical contribution is reflected in the following lemma, which bounds the bonuses' expected value.

**Lemma 4.2.** *With probability at least $1 - \delta/4$,*

$$\mathbb{E}_c\left[ \sum_{t=2}^{T} \sum_{h,s,a} q_h(s,a|\pi_c^t, P_c^\star) b_c^t(s,a) \right] \lesssim$$

$$\frac{TH^7 \log(TH|\mathcal{P}|/\delta)}{\min\{\beta_P, \beta_\ell\}} + (\beta_\ell + 2H\beta_P)|S||A| \log(T+1).$$

For continuity of the analysis, we defer the proof sketch to Section 4.2. Complete statement and full proof appear in Lemma A.11. We note, however, that for our specific choices of $\beta_\ell$ and $\beta_P$ given in the proof of Theorem A.12, the expected sum of the bonus terms is bounded by $\widetilde{O}(\sqrt{TH^8|S||A| \log(|\mathcal{F}||\mathcal{P}|/\delta)})$. This bound is independent of any additional dimensional factors and matches the standard non-contextual bonus contribution to the regret in terms of $T$, $S$, and $A$ (see, e.g., Jaksch et al., 2010).

Next, we keep decomposing $(ii)$. Using the value difference lemma (Lemma B.4) we obtain,

$$(ii) = \underbrace{\sum_{t=1}^{T} \mathbb{E}_{c_t}\left[ \sum_{h=1}^{H} \Lambda_{c_t,h}^t \right]}_{(iii)} + \underbrace{\sum_{t=1}^{T} \mathbb{E}_{c_t}\left[ \sum_{h=1}^{H} \Phi_{c_t,h}^t \right]}_{(iv)},$$

Where $\Lambda_{c_t,h}^t, \Phi_{c_t,h}^t$ are the standard value difference terms:

$$\Lambda_{c_t,h}^t = \mathbb{E}_{\pi_{c_t}^\star, P_{c_t}^\star}\left[ \left\langle \hat{Q}_{c_t,h}^t(s_h^t, \cdot), \pi_{c_t}^t(\cdot|s_h^t) - \pi_{c_t}^\star(\cdot|s_h^t) \right\rangle | s_1 \right],$$

$$\Phi^t_{c_t,h} = \mathbb{E}_{\pi^\star_{c_t}, P^\star_{c_t}} \Big[ \hat{Q}^t_{c_t,h}(s^t_h, a^t_h) - \ell^\star_{c_t}(s^t_h, a^t_h)$$
$$- P^\star_{c_t} \hat{V}^t_{c_t,h+1}(s^t_h, a^t_h) \mid s_1 \Big].$$

The bound of term $(iii)$ directly follows from the fundamental inequality of Online Mirror Descent (OMD; see Appendix B.1 for more information) algorithm, for an appropriate learning rate choice, as stated next. Full proof appears in Lemma A.2.

**Lemma 4.3.** *For $\eta = \sqrt{2\log|A|/H^2 T}$, it holds true that*

$$(iii) \le \sqrt{2H^4 T \log|A|}.$$

We then bound term $(iv)$. The bound follows from the optimistic design of the bonuses, which makes them vanish in this case. The resulting asymptotic bound is given below, and the full statement and proof appear in Lemma A.3.

**Lemma 4.4.** *With probability at least $1 - \delta/4$,*

$$(iv) \lesssim \left( \frac{1}{\beta_\ell} + \frac{H}{\beta_P} \right) TH^3 \log(TH|\mathcal{F}||\mathcal{P}|/\delta).$$

Finally, we conclude our final regret bound by combining the results above and choosing the parameters $\beta_\ell \approx \sqrt{\frac{TH^7 \log(TH|\mathcal{F}||\mathcal{P}|/\delta)}{|S||A|\log(T+1)}} \approx \sqrt{H}\beta_P$. Complete proof is provided in Theorem A.12.

### 4.2. Main Technical Novelty: Deriving the Bonus Bound

The final and most technically significant component of our analysis is the derivation of the bonus bound. Our key idea is to bound the bonus separately for each context and only then take expectation over the context, enabling the use of the dynamics generalization guarantee, in contrast to prior works that typically reverse this order. For a fixed context, we analyze four events describing the quality of the dynamics estimation and derive corresponding upper bounds on the bonus. The technical details are given below, yielding the proof of Lemma 4.2.

Fix any context $c \in \mathcal{C}$ and let $\gamma, \gamma' > 0$ satisfy $H^4 \gamma \lesssim \gamma' \le \frac{1}{2} \min\{\beta_P, \beta_\ell\}$. These parameters define two classes of events describing the quality of the learned dynamics for context $c$, as stated next.

**Event Type A:** *Expected dynamics approximation error.* For any $t \ge 2$, let $A^t_c$ be the event that the expected dynamics approximation error (measured in squared Hellinger distance) at round $t$ for the context $c$ exceeds $\gamma$. Intuitively, $A^t_c$ captures the bad cases where the learned dynamics at time $t$, $\hat{P}^t_c$ is an inaccurate approximation of $P^c_\star$. Formally,

$$A^t_c = \left\{ \sum_{i=1}^{t-1} \mathbb{E}_{\pi^i_c, P^\star_c} \left[ \sum_{h=1}^{H} D^2_H(\hat{P}^t_c(\cdot|s^i_h, a^i_h), P^\star_c(\cdot|s^i_h, a^i_h)) \right] > \gamma \right\}$$

Its complement event, $\bar{A}^t_c$, indicates that the expected approximation error is at most $\gamma$, meaning $P^c_\star$ is well approximated by $\hat{P}^t_c$.

**Event Type B:** *Expected visitations.* For each $t \ge 2$, layer $h \in [H]$, and state-action pair $(s, a) \in S_h \times A$, let $B^t_c(h, s, a)$ define the event that the expected number of visits to $(s, a)$ under context $c$ up to round $t - 1$ is at most $\gamma' - 1$. Intuitively, $B^t_c(h, s, a)$ identifies counterfactually under-explored state-action pairs of $P^\star_c$. Formally,

$$B^t_c(h, s, a) = \left\{ 1 + \sum_{i=1}^{t-1} q_h(s, a|\pi^i_c, P^\star_c) \le \gamma' \right\}.$$

The complementary event $\bar{B}^t_c(h, s, a)$ indicates that the expected number of visits exceeds $\gamma' - 1$, meaning it is sufficient for learning $P^\star_c(\cdot|s, a)$, in expectation.

These two types of events jointly quantify how well the dynamics have been approximated relative to the counterfactual expected number of samples collected for each context-state-action triplet up to time $t$. Together, they characterize the amount of exploration still required to refine the dynamics model before reliable exploitation becomes possible, affecting the cumulative cost of the bonuses. We identify that under a complementary combination of these events, we conclude the useful properties elaborated next. These properties allow us to bound the sum of bonuses in each case. The first property is simply implied by Markov inequality ($\mathbb{I}[E]$ denotes an indicator that event $E$ holds).

**Observation 4.5.** *For any $t \ge 2$,*

$$\mathbb{I}[A^t_c] \le \frac{1}{\gamma} \sum_{i=1}^{t-1} \mathbb{E}_{\pi^i_c, P^\star_c} \left[ \sum_{h=1}^{H} D^2_H(\hat{P}^t_c(\cdot|s^i_h, a^i_h), P^\star_c(\cdot|s^i_h, a^i_h)) \right].$$

For the second property, we observe that for any $(s, a) \in S_h \times A$ that are underexplored at round $t$, (meaning, $B^t_c(h, s, a)$ holds), the bonus has maximal value of 1. By $\gamma'$ choice, this immediately implies the bonuses are also bounded by the true occupancy measures, meaning by

$$\frac{\beta/2}{1 + \sum_{i=1}^{t-1} q_h(s, a|\pi^i_c, P^\star_c)}.$$

This is informally given in the next claim. The accurate statement and proof are given in Claim A.6 (Appendix A.2).

**Claim 4.6.** *For all $t \ge 2$, $h \in [H]$ and $(s, a) \in S_h \times A$ such that event $B^t_c(h, s, a)$ holds, we have for $\beta \in \{\beta_P, \beta_\ell\}$,*

$$b^t_\beta(c, s, a) \le \frac{\beta/2}{1 + \sum_{i=1}^{t-1} q_h(s, a|\pi^i_c, P^\star_c)}.$$

Lastly, but most importantly, we observe that for tuples $(c, t, h, s, a)$ for which both good events $\bar{A}^t_c$ and $\bar{B}^t_c(h, s, a)$

hold, the approximated and true occupancy measures are different by a small constant multiplicative factor. The intuition behind this observation is that for such tuples both the expected approximation error of the dynamics is small and they are counterfactually visited sufficient number of times in expectation. Hence, their related occupancy measures are adequately studied. This is informally given in the next lemma. Full statement and proof are given in Lemma A.7.

**Lemma 4.7.** *Let any $t \geq 2$, $h \in [H]$, $(s,a) \in S_h \times A$ such that both events $\bar{A}_c^t$ and $\bar{B}_c^t(h,s,a)$ hold. Then,*

$$1 + \sum_{i=1}^{t-1} q_h(s,a|\pi_c^i, \hat{P}_c^t) \geq \frac{1}{6}\left(1 + \sum_{i=1}^{t-1} q_h(s,a|\pi_c^i, P_c^\star)\right).$$

To use the properties given in Observation 4.5, Claim 4.6, and Lemma 4.7, we split the cumulative bonus sum of the context $c$ into three sums by the complementary combination of the four event types as given in Appendix A.2, Equation (6). The intuition behind this decomposition is based on the three properties above, and give below.

For time steps $t \in [2, T]$ for which event $A_c^t$ holds, although the specific dynamics approximation error is high, the cumulative expected error in squared Hellinger distance is logarithmic. Hence, we can control this quantity using $\gamma \approx \sqrt{T}$, as shown in Observation 4.5. This is informally given next. For full statement and proof, see Lemma A.8.

**Lemma 4.8.** *The following holds true.*

$$\sum_{t=2}^{T} \mathbb{I}\big[A_c^t\big] \mathbb{E}_{\pi_c^t, P_c^\star}\left[\sum_{h=1}^{H} b_c^t(s_h^t, a_h^t)\right] \lesssim$$
$$\frac{H^2}{\gamma} \sum_{t=2}^{T} \sum_{i=1}^{t-1} \mathbb{E}_{\pi_c^i, P_c^\star}\left[\sum_{h=1}^{H} D_H^2(\hat{P}_c^t(\cdot|s_h^i, a_h^i), P_c^\star(\cdot|s_h^i, a_h^i))| s_1\right]$$

Next, time steps for which the complementary event $\bar{A}_c^t$ holds, we split into two additional sums, according to tuples $(h,s,a) \in [H] \times S_h \times A$ for which event $B_c^t(h,s,a)$ holds and those $\bar{B}_c^t(h,s,a)$ holds.

To bound the first-mentioned sum, we use Claim 4.6. It then allows us to replace the bonus with the true occupancy measures. Then, the sum of inverse occupancy measures is bounded by $O(|S||A|\log(T+1))$. This is given in the next lemma. Formal statement and proof appear in Lemma A.9.

**Lemma 4.9.** *It holds that,*

$$\sum_{t=2}^{T} \mathbb{E}_{\pi_c^t, P_c^\star}\left[\sum_{h=1}^{H} \mathbb{I}\big[\bar{A}_c^t, B_c^t(h, s_h^t, a_h^t)\big] b_c^t(s_h^t, a_h^t)\right] \leq$$
$$|S||A|\log(T+1)(\beta_\ell + 2H\beta_P).$$

Lastly, we bound the expected bonuses sum under both complementary events. For this case, we apply Lemma 4.7,

which allows us to again replace the bonuses with the true occupancy measures, as given next. See Lemma A.10 for formal statement and proof.

**Lemma 4.10.** *The following holds true.*

$$\sum_{t=2}^{T} \mathbb{E}_{\pi_c^t, P_c^\star}\left[\sum_{h=1}^{H} \mathbb{I}\big[\bar{A}_c^t, \bar{B}_c^t(h, s_h^t, a_h^t)\big] b_c^t(s_h^t, a_h^t)\right] \leq$$
$$6(\beta_\ell + 2H\beta_P)|S||A|\log(T+1).$$

*Proof of Lemma 4.2.* To conclude Lemma 4.2 we combine together all the results of Lemmas 4.8 to 4.10 and take an expectation over the context. We are then able to bound the sums of expected squared Hellinger errors with $O(T \log |\mathcal{P}|)$ (using Corollary 2.3). which, for $\gamma = O(\sqrt{T})$ yields the desired bound. □

For full proof, see Lemma A.11 in the appendix.

## 5. Discussion and Conclusion

In this work, we presented `OPO-CMDP`, a new policy optimization algorithm for stochastic CMDPs that uses general offline function approximation. `OPO-CMDP` integrates optimism-based exploration with exponential-weighting policy updates, offering a simple and efficient method for regret minimization. Our analysis established that `OPO-CMDP` achieves a near-optimal regret of $\widetilde{O}(H^4\sqrt{T|S||A|\log(|\mathcal{F}||\mathcal{P}|)})$, which improves the dependence on $|S|$ and $|A|$ compared to previous works and is optimal up to $H^{3.5}$. This bound demonstrates that near-optimal regret is achievable under general function approximation without requiring strong structural assumptions or a complicated algorithmic approach.

Our main technical contribution is the derivation of new exploration bonuses for offline function approximation that require no additional assumptions on the function class or the CMDP.

Promising directions for future work include improving the dependence on the horizon $H$ to achieve optimality and eliminating the need for counterfactual occupancy measure computation, both of which present non-trivial technical challenges. At a broader level, it would be of interest to extend our approach to larger-scale reinforcement learning settings with small but latent state spaces, such as block MDPs or MDPs with rich observations.

## Impact Statement

This paper presents work whose goal is to advance the field of Machine Learning. There are many potential societal consequences of our work, none which we feel must be specifically highlighted here.

## Acknowledgments

This project has received funding from the European Research Council (ERC) under the European Union's Horizon 2020 research and innovation program (grant agreement No. 882396), by the Israel Science Foundation, the Yandex Initiative for Machine Learning at Tel Aviv University and a grant from the Tel Aviv University Center for AI and Data Science (TAD).

OL is also supported by the Google PhD Fellowship Award (2025).

AC is supported by the Israel Science Foundation (ISF, grant no. 2250/22).

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

# A. Proofs

## A.1. Deriving The Value-Confidence Bounds

**Lemma A.1** (Restatement of Lemma 4.1). *With probability at least $1 - \delta/4$,*

$$(i) = \sum_{t=1}^{T} \mathbb{E}_{c_t}\left[V_{c_t}^{\pi_{c_t}^t}(s_1) - \hat{V}_{c_t}^t(s_1)\right]$$

$$\leq H + \frac{112TH^3 \log(128T^4 H|\mathcal{F}||\mathcal{P}|/\delta^2)}{\beta_\ell} + \frac{25TH^4 \log(8TH|\mathcal{P}|/\delta)}{\beta_P}$$

$$+ 2\mathbb{E}_c\left[\sum_{t=2}^{T}\sum_{h=1}^{H}\sum_{s\in S_h}\sum_{a\in A} q_h(s,a|\pi_c^t, P_c^\star)\left(b_{\beta_\ell}^t(c,s,a) + 2Hb_{\beta_P}^t(c,s,a)\right)\right].$$

*Proof.*

$$(i) = \mathbb{E}_c\left[\sum_{t=1}^{T} V_c^{\pi_c^t}(s_1) - \hat{V}_c^t(s_1)\right] \qquad \text{(Linearity of expectation)}$$

$$\leq H + \mathbb{E}_c\left[\sum_{t=2}^{T} V_c^{\pi_c^t}(s_1) - \hat{V}_c^t(s_1)\right]$$

(Since both values are bounded in $[0, H]$, the total difference in a single round is upper bounded by $H$.)

$$= H + \mathbb{E}_c\left[\sum_{t=2}^{T} \mathbb{E}_{\pi_c^t, P_c^\star}\left[\sum_{h=1}^{H} \ell_c^\star(s_h^t, a_h^t) + P_c^\star \hat{V}_{c,h+1}^t(s_h^t, a_h^t) - \hat{Q}_{c,h+1}^t(s_h^t, a_h^t) \mid s_1\right]\right]$$

(By the extended value-difference lemma, opening by $P^\star$, see Lemma B.4)

$$= H + \mathbb{E}_c\left[\sum_{t=2}^{T} \mathbb{E}_{\pi_c^t, P_c^\star}\left[\sum_{h=1}^{H} \ell_c^\star(s_h^t, a_h^t) + P_c^\star \hat{V}_{c,h+1}^t(s_h^t, a_h^t) \mid s_1\right]\right] \qquad \text{(By $\hat{Q}_{h+1}^t$ definition)}$$

$$- \mathbb{E}_c\left[\sum_{t=2}^{T} \mathbb{E}_{\pi_c^t, P_c^\star}\left[\sum_{h=1}^{H} \max\left\{\hat{\ell}_c^t(s_h^t, a_h^t) + \hat{P}_c^t \hat{V}_{c,h+1}^t(s_h^t, a_h^t), 0\right\} \mid s_1\right]\right]$$

$$\leq H + \mathbb{E}_c\left[\sum_{t=2}^{T} \mathbb{E}_{\pi_c^t, P_c^\star}\left[\sum_{h=1}^{H} \ell_c^\star(s_h^t, a_h^t) + P_c^\star \hat{V}_{c,h+1}^t(s_h^t, a_h^t) \mid s_1\right]\right] \qquad \text{(For all $a, b \in \mathbb{R}, -\max\{a,b\} \leq -a$.)}$$

$$- \mathbb{E}_c\left[\sum_{t=2}^{T} \mathbb{E}_{\pi_c^t, P_c^\star}\left[\sum_{h=1}^{H} \hat{\ell}_c^t(s_h^t, a_h^t) + \hat{P}_c^t \hat{V}_{c,h+1}^t(s_h^t, a_h^t) \mid s_1\right]\right]$$

$$= H + \mathbb{E}_c\left[\sum_{t=2}^{T} \mathbb{E}_{\pi_c^t, P_c^\star}\left[\sum_{h=1}^{H} \ell_c^\star(s_h^t, a_h^t) - \hat{\ell}_c^t(s_h^t, a_h^t) \mid s_1\right]\right] \qquad \text{(Rearrange terms)}$$

$$+ \mathbb{E}_c\left[\sum_{t=2}^{T} \mathbb{E}_{\pi_c^t, P_c^\star}\left[\sum_{h=1}^{H} \left(P_c^\star(\cdot|s_h^t, a_h^t) - \hat{P}_c^t(\cdot|s_h^t, a_h^t)\right)\hat{V}_{c,h+1}^t(\cdot) \mid s_1\right]\right]$$

$$\leq H + \mathbb{E}_c\left[\sum_{t=2}^{T} \mathbb{E}_{\pi_c^t, P_c^\star}\left[\sum_{h=1}^{H} \ell_c^\star(s_h^t, a_h^t) - \hat{\ell}_c^t(s_h^t, a_h^t) \mid s_1\right]\right] \qquad \text{(Since $|\hat{V}_{c,h}^t(s)| \leq H, \ \forall c, t, h, s$.)}$$

$$+ H\mathbb{E}_c\left[\sum_{t=2}^{T} \mathbb{E}_{\pi_c^t, P_c^\star}\left[\sum_{h=1}^{H} \|P_c^\star(\cdot|s_h^t, a_h^t) - \hat{P}_c^t(\cdot|s_h^t, a_h^t)\|_1 \mid s_1\right]\right]$$

$$= H + \mathbb{E}_c\left[\sum_{t=2}^{T}\sum_{h=1}^{H}\sum_{s\in S_h}\sum_{a\in A} q_h(s, a|\pi_c^t, P_c^\star)\left(\ell_c^\star(s, a) - \hat{\ell}_c^t(s, a)\right)\right]$$

(Re-write using occupancy measures w.r.t $\pi^t, P^\star$)

$$+ H\mathbb{E}_c\left[\sum_{t=2}^{T}\sum_{h=1}^{H}\sum_{s\in S_h}\sum_{a\in A} q_h(s,a|\pi_c^t, P_c^\star)\|P_c^\star(\cdot|s,a) - \hat{P}_c^t(\cdot|s,a)\|_1\right]$$

$$= H + \mathbb{E}_c\left[\sum_{t=2}^{T}\sum_{h=1}^{H}\sum_{s\in S_h}\sum_{a\in A} q_h(s,a|\pi_c^t, P_c^\star)\Big(f_c^\star(s,a) - \hat{f}_c^t(s,a) + b_{\beta_\ell}^t(c,s,a) + 2Hb_{\beta_P}^t(c,s,a)\Big)\right]$$
$$\text{(By } \ell^\star, \hat{\ell}^t \text{ definitions.)}$$

$$+ H\mathbb{E}_c\left[\sum_{t=2}^{T}\sum_{h=1}^{H}\sum_{s\in S_h}\sum_{a\in A} q_h(s,a|\pi_c^t, P_c^\star)\cdot 2\cdot\frac{1}{2}\|P_c^\star(\cdot|s,a) - \hat{P}_c^t(\cdot|s,a)\|_1\right]$$

$$= H + \mathbb{E}_c\left[\sum_{t=2}^{T}\sum_{h=1}^{H}\sum_{s\in S_h}\sum_{a\in A} q_h(s,a|\pi_c^t, P_c^\star)\Big(f_c^\star(s,a) - \hat{f}_c^t(s,a)\Big)\right]$$

$$+ 2H\mathbb{E}_c\left[\sum_{t=2}^{T}\sum_{h=1}^{H}\sum_{s\in S_h}\sum_{a\in A} q_h(s,a|\pi_c^t, P_c^\star)\cdot\frac{1}{2}\|P_c^\star(\cdot|s,a) - \hat{P}_c^t(\cdot|s,a)\|_1\right]$$

$$+ \mathbb{E}_c\left[\sum_{t=2}^{T}\sum_{h=1}^{H}\sum_{s\in S_h}\sum_{a\in A} q_h(s,a|\pi_c^t, P_c^\star)\big(b_{\beta_\ell}^t(c,s,a) + 2Hb_{\beta_P}^t(c,s,a)\big)\right]$$

$$\leq H + \mathbb{E}_c\left[\sum_{t=2}^{T}\sum_{h=1}^{H}\sum_{s\in S_h}\sum_{a\in A} q_h(s,a|\pi_c^t, P_c^\star)|f_c^\star(s,a) - \hat{f}_c^t(s,a)|\right]$$

$$+ 2H\mathbb{E}_c\left[\sum_{t=2}^{T}\sum_{h=1}^{H}\sum_{s\in S_h}\sum_{a\in A} q_h(s,a|\pi_c^t, P_c^\star)\cdot\frac{1}{2}\|P_c^\star(\cdot|s,a) - \hat{P}_c^t(\cdot|s,a)\|_1\right]$$

$$+ \mathbb{E}_c\left[\sum_{t=2}^{T}\sum_{h=1}^{H}\sum_{s\in S_h}\sum_{a\in A} q_h(s,a|\pi_c^t, P_c^\star)\big(b_{\beta_\ell}^t(c,s,a) + 2Hb_{\beta_P}^t(c,s,a)\big)\right]$$

$$= H + \mathbb{E}_c\left[\sum_{t=2}^{T}\sum_{h=1}^{H}\sum_{s\in S_h}\sum_{a\in A} q_h(s,a|\pi_c^t, P_c^\star)\min\Big\{1, |f_c^\star(s,a) - \hat{f}_c^t(s,a)|\Big\}\right] \quad \text{(Since all values are bounded in } [0,1])$$

$$+ 2H\mathbb{E}_c\left[\sum_{t=2}^{T}\sum_{h=1}^{H}\sum_{s\in S_h}\sum_{a\in A} q_h(s,a|\pi_c^t, P_c^\star)\min\Big\{1, \frac{1}{2}\|P_c^\star(\cdot|s,a) - \hat{P}_c^t(\cdot|s,a)\|_1\Big\}\right]$$

$$+ \mathbb{E}_c\left[\sum_{t=2}^{T}\sum_{h=1}^{H}\sum_{s\in S_h}\sum_{a\in A} q_h(s,a|\pi_c^t, P_c^\star)\big(b_{\beta_\ell}^t(c,s,a) + 2Hb_{\beta_P}^t(c,s,a)\big)\right]$$

$$= H + \mathbb{E}_c\left[\sum_{t=2}^{T}\sum_{h,s,a} q_h(s,a|\pi_c^t, P_c^\star)\min\left\{1, \sqrt{\frac{\beta_\ell}{\beta_\ell}\frac{1+\sum_{i=1}^{t-1}q_h(s,a|\pi_c^i, \hat{P}_c^t)}{1+\sum_{i=1}^{t-1}q_h(s,a|\pi_c^i, \hat{P}_c^t)}}|f_c^\star(s,a) - \hat{f}_c^t(s,a)|\right\}\right]$$
$$\text{(As shorthand, we denote } \sum_{h=1}^{H}\sum_{s\in S_h}\sum_{a\in A} \text{ as } \sum_{h,s,a})$$

$$+ 2H\mathbb{E}_c\left[\sum_{t=2}^{T}\sum_{h,s,a} q_h(s,a|\pi^t(c,), P_c^\star)\min\left\{1, \sqrt{\frac{\beta_P}{\beta_P}\frac{1+\sum_{i=1}^{t-1}q_h(s,a|\pi_c^i, \hat{P}_c^t)}{1+\sum_{i=1}^{t-1}q_h(s,a|\pi_c^i, \hat{P}_c^t)}}\frac{1}{2}\|P_c^\star(\cdot|s,a) - \hat{P}_c^t(\cdot|s,a)\|_1\right\}\right]$$

$$+ \mathbb{E}_c\left[\sum_{t=2}^{T}\sum_{h=1}^{H}\sum_{s\in S_h}\sum_{a\in A} q_h(s,a|\pi_c^t, P_c^\star)\big(b_{\beta_\ell}^t(c,s,a) + 2Hb_{\beta_P}^t(c,s,a)\big)\right]$$

$$\leq H + \mathbb{E}_c\Big[\sum_{t=2}^{T}\sum_{h,s,a} q_h(s,a|\pi_c^t, P_c^\star)\min\Big\{1, \frac{\beta_\ell/2}{1+\sum_{i=1}^{t-1}q_h(s,a|\pi_c^i, \hat{P}_c^t)} \qquad\qquad \text{(By AM-GM)}$$

$$+ \frac{1}{2\beta_\ell}\bigg(1+\sum_{i=1}^{t-1}q_h(s,a|\pi_c^i, \hat{P}_c^t)\bigg)\Big(f_c^\star(s,a) - \hat{f}_c^t(s,a)\Big)^2\Big\}\Big]$$

$$+ 2H\mathbb{E}_c\Big[\sum_{t=2}^{T}\sum_{h,s,a}q_h(s,a|\pi^t(c,),P_c^\star)\min\Big\{1,\frac{\beta_P/2}{1+\sum_{i=1}^{t-1}q_h(s,a|\pi_c^i,\hat{P}_c^t)}$$

$$+ \frac{1}{2\beta_P}\Big(1+\sum_{i=1}^{t-1}q_h(s,a|\pi_c^i,\hat{P}_c^t)\Big)\frac{1}{4}\|P_c^\star(\cdot|s,a)-\hat{P}_c^t(\cdot|s,a)\|_1^2\Big\}\Big]$$

$$+ \mathbb{E}_c\Big[\sum_{t=2}^{T}\sum_{h=1}^{H}\sum_{s\in S_h}\sum_{a\in A}q_h(s,a|\pi_c^t,P_c^\star)\big(b_{\beta_\ell}^t(c,s,a)+2Hb_{\beta_P}^t(c,s,a)\big)\Big]$$

$$\leq H + \mathbb{E}_c\Big[\sum_{t=2}^{T}\sum_{h,s,a}q_h(s,a|\pi_c^t,P_c^\star)\Big(\min\Big\{1,\frac{\beta_\ell/2}{1+\sum_{i=1}^{t-1}q_h(s,a|\pi_c^i,\hat{P}_c^t)}\Big\} \qquad \text{(By min properties)}$$

$$+ \frac{1}{2\beta_\ell}\Big(1+\sum_{i=1}^{t-1}q_h(s,a|\pi_c^i,\hat{P}_c^t)\Big)\Big(f_c^\star(s,a)-\hat{f}_c^t(s,a)\Big)^2\Big)\Big]$$

$$+ 2H\mathbb{E}_c\Big[\sum_{t=2}^{T}\sum_{h,s,a}q_h(s,a|\pi^t(c,),P_c^\star)\Big(\min\Big\{1,\frac{\beta_P/2}{1+\sum_{i=1}^{t-1}q_h(s,a|\pi_c^i,\hat{P}_c^t)}\Big\}$$

$$+ \frac{1}{2\beta_P}\Big(1+\sum_{i=1}^{t-1}q_h(s,a|\pi_c^i,\hat{P}_c^t)\Big)\frac{1}{4}\|P_c^\star(\cdot|s,a)-\hat{P}_c^t(\cdot|s,a)\|_1^2\Big)\Big]$$

$$+ \mathbb{E}_c\Big[\sum_{t=2}^{T}\sum_{h=1}^{H}\sum_{s\in S_h}\sum_{a\in A}q_h(s,a|\pi_c^t,P_c^\star)\big(b_{\beta_\ell}^t(c,s,a)+2Hb_{\beta_P}^t(c,s,a)\big)\Big]$$

$$\leq H + \frac{TH}{2\beta_\ell} + \frac{1}{2\beta_\ell}\mathbb{E}_c\Big[\sum_{t=2}^{T}\sum_{i=1}^{t-1}\sum_{h=1}^{H}\sum_{s\in S_h}\sum_{a\in A}q_h(s,a|\pi_c^i,\hat{P}_c^t)\Big(\hat{f}_c^t(s,a)-f_c^\star(s,a)\Big)^2\Big]$$

$$\text{(Re-ordering, using that }\big(f^\star-\hat{f}^t\big)^2\leq 1, \|P^\star-\hat{P}^t\|_1^2\leq 4 \text{ and the bonuses definition)}$$

$$+ \frac{TH^2}{\beta_P} + \frac{H}{\beta_P}\mathbb{E}_c\Big[\sum_{t=2}^{T}\sum_{i=1}^{t-1}\sum_{h=1}^{H}\sum_{s\in S_h}\sum_{a\in A}q_h(s,a|\pi_c^i,\hat{P}_c^t)\frac{1}{4}\|\hat{P}_c^t(\cdot|s,a)-P_c^\star(\cdot|s,a)\|_1^2\Big]$$

$$+ 2\mathbb{E}_c\Big[\sum_{t=2}^{T}\sum_{h=1}^{H}\sum_{s\in S_h}\sum_{a\in A}q_h(s,a|\pi_c^t,P_c^\star)\big(b_{\beta_\ell}^t(c,s,a)+2Hb_{\beta_P}^t(c,s,a)\big)\Big]$$

$$= H + \frac{TH}{2\beta_\ell} + \frac{1}{2\beta_\ell}\mathbb{E}_c\Big[\sum_{t=2}^{T}\sum_{i=1}^{t-1}\mathbb{E}_{\pi_c^i,\hat{P}_c^t}\Big[\sum_{h=1}^{H}\Big(\hat{f}_c^t(s_h^i,a_h^i)-f_c^\star(s_h^i,a_h^i)\Big)^2\big|s_1\Big]\Big]$$

$$\text{(Translate occupancy measures to expectations)}$$

$$+ \frac{TH^2}{\beta_P} + \frac{H}{\beta_P}\mathbb{E}_c\Big[\sum_{t=2}^{T}\sum_{i=1}^{t-1}\mathbb{E}_{\pi_c^i,\hat{P}_c^t}\Big[\sum_{h=1}^{H}\frac{1}{4}\|\hat{P}_c^t(\cdot|s_h^i,a_h^i)-P_c^\star(\cdot|s_h^i,a_h^i)\|_1^2\big|s_1\Big]\Big]$$

$$+ 2\mathbb{E}_c\Big[\sum_{t=2}^{T}\sum_{h=1}^{H}\sum_{s\in S_h}\sum_{a\in A}q_h(s,a|\pi_c^t,P_c^\star)\big(b_{\beta_\ell}^t(c,s,a)+2Hb_{\beta_P}^t(c,s,a)\big)\Big]$$

$$\leq H + \frac{TH}{2\beta_\ell} + \frac{3}{2\beta_\ell}\mathbb{E}_c\Big[\sum_{t=2}^{T}\sum_{i=1}^{t-1}\mathbb{E}_{\pi_c^i,P_c^\star}\Big[\sum_{h=1}^{H}\Big(\hat{f}_c^t(s_h^i,a_h^i)-f_c^\star(s_h^i,a_h^i)\Big)^2\big|s_1\Big]\Big]$$

$$\text{(By the value change of measure Lemma B.2)}$$

$$+ \frac{9H^2}{2\beta_\ell}\mathbb{E}_c\Big[\sum_{t=2}^{T}\sum_{i=1}^{t-1}\mathbb{E}_{\pi_c^i,P_c^\star}\Big[\sum_{h=1}^{H}D_H^2\Big(\hat{P}_c^t(\cdot|s_h^i,a_h^i),P_c^\star(\cdot|s_h^i,a_h^i)\Big)\mid s_1\Big]\Big]$$

$$+ \frac{TH^2}{\beta_P} + \frac{3H}{\beta_P}\mathbb{E}_c\Big[\sum_{t=2}^{T}\sum_{i=1}^{t-1}\mathbb{E}_{\pi_c^i,P_c^\star}\Big[\sum_{h=1}^{H}\frac{1}{4}\|\hat{P}_c^t(\cdot|s_h^i,a_h^i)-P_c^\star(\cdot|s_h^i,a_h^i)\|_1^2\big|s_1\Big]\Big]$$

$$+ \frac{9H^3}{\beta_P} \mathbb{E}_c \left[ \sum_{t=2}^{T} \sum_{i=1}^{t-1} \mathbb{E}_{\pi_c^i, P_c^\star} \left[ \sum_{h=1}^{H} D_H^2 \left( \hat{P}_c^t(\cdot|s_h^i, a_h^i), P_c^\star(\cdot|s_h^i, a_h^i) \right) \mid s_1 \right] \right]$$

$$+ 2\mathbb{E}_c \left[ \sum_{t=2}^{T} \sum_{h=1}^{H} \sum_{s \in S_h} \sum_{a \in A} q_h(s, a | \pi_c^t, P_c^\star) \left( b_{\beta_\ell}^t(c, s, a) + 2H b_{\beta_P}^t(c, s, a) \right) \right]$$

$$\leq H + \frac{TH}{2\beta_\ell} + \frac{TH^2}{\beta_P} + \frac{3}{2\beta_\ell} \mathbb{E}_c \left[ \sum_{t=2}^{T} \sum_{i=1}^{t-1} \mathbb{E}_{\pi_c^i, P_c^\star} \left[ \sum_{h=1}^{H} \left( \hat{f}_c^t(s_h^i, a_h^i) - f_c^\star(s_h^i, a_h^i) \right)^2 \mid s_1 \right] \right]$$

(For any two distributions $Q, P$, $\|P - Q\|_1^2 \leq 4D_H^2(P, Q)$)

$$+ \left( \frac{9H^2}{2\beta_\ell} + \frac{9H^3}{\beta_P} + \frac{3H}{\beta_P} \right) \mathbb{E}_c \left[ \sum_{t=2}^{T} \sum_{i=1}^{t-1} \mathbb{E}_{\pi_c^i, P_c^\star} \left[ \sum_{h=1}^{H} D_H^2 \left( \hat{P}_c^t(\cdot|s_h^i, a_h^i), P_c^\star(\cdot|s_h^i, a_h^i) \right) \mid s_1 \right] \right]$$

$$+ 2\mathbb{E}_c \left[ \sum_{t=2}^{T} \sum_{h=1}^{H} \sum_{s \in S_h} \sum_{a \in A} q_h(s, a | \pi_c^t, P_c^\star) \left( b_{\beta_\ell}^t(c, s, a) + 2H b_{\beta_P}^t(c, s, a) \right) \right]$$

$$\leq H + \frac{TH}{2\beta_\ell} + \frac{TH^2}{\beta_P} + \frac{3}{2\beta_\ell} \sum_{t=2}^{T} \mathbb{E}_c \left[ \sum_{i=1}^{t-1} \mathbb{E}_{\pi_c^i, P_c^\star} \left[ \sum_{h=1}^{H} \left( \hat{f}_c^t(s_h^i, a_h^i) - f_c^\star(s_h^i, a_h^i) \right)^2 \right] \right]$$

($s_1$ is identical for all contexts so that we can omit the condition, then apply linearity of expectation)

$$+ \left( \frac{9H^2}{2\beta_\ell} + \frac{9H^3}{\beta_P} + \frac{3H}{\beta_P} \right) \sum_{t=2}^{T} \mathbb{E}_c \left[ \sum_{i=1}^{t-1} \mathbb{E}_{\pi_c^i, P_c^\star} \left[ \sum_{h=1}^{H} D_H^2 \left( \hat{P}_c^t(\cdot|s_h^i, a_h^i), P_c^\star(\cdot|s_h^i, a_h^i) \right) \right] \right]$$

$$+ 2\mathbb{E}_c \left[ \sum_{t=2}^{T} \sum_{h=1}^{H} \sum_{s \in S_h} \sum_{a \in A} q_h(s, a | \pi_c^t, P_c^\star) \left( b_{\beta_\ell}^t(c, s, a) + 2H b_{\beta_P}^t(c, s, a) \right) \right]$$

$$\leq H + \frac{TH}{2\beta_\ell} + \frac{TH^2}{\beta_P} + \frac{3T}{2\beta_\ell} 68H \log(16T^3|\mathcal{F}|/\delta) + \left( \frac{9H^2}{2\beta_\ell} + \frac{9H^3}{\beta_P} + \frac{3H}{\beta_P} \right) T2H \log(8TH|\mathcal{P}|/\delta)$$

(By union bound over Corollaries 2.1 and 2.3, holds with probability at least $1 - \delta/4$)

$$+ 2\mathbb{E}_c \left[ \sum_{t=2}^{T} \sum_{h=1}^{H} \sum_{s \in S_h} \sum_{a \in A} q_h(s, a | \pi_c^t, P_c^\star) \left( b_{\beta_\ell}^t(c, s, a) + 2H b_{\beta_P}^t(c, s, a) \right) \right]$$

$$\leq H + \frac{112TH^3 \log(128T^4H|\mathcal{F}||\mathcal{P}|/\delta^2)}{\beta_\ell} + \frac{25TH^4 \log(8TH|\mathcal{P}|/\delta)}{\beta_P}$$

$$+ 2\mathbb{E}_c \left[ \sum_{t=2}^{T} \sum_{h=1}^{H} \sum_{s \in S_h} \sum_{a \in A} q_h(s, a | \pi_c^t, P_c^\star) \left( b_{\beta_\ell}^t(c, s, a) + 2H b_{\beta_P}^t(c, s, a) \right) \right],$$

as stated. $\qquad\square$

**Lemma A.2** (Restatement of Lemma 4.3). *For $\eta = \sqrt{2 \log|A|/H^2 T}$, the following holds true.*

$$(iii) \leq \sqrt{2H^4 T \log|A|}.$$

*Proof.* We open the proof by observing that

$$(iii) = \sum_{t=1}^{T} \mathbb{E}_{c_t} \left[ \sum_{h=1}^{H} \mathbb{E}_{\pi_{c_t}^\star, P_{c_t}^\star} \left[ \left\langle \hat{Q}_{c_t, h}^t(s_h^t, \cdot), \pi_{c_t}^t(\cdot|s_h^t) - \pi_{c_t}^\star(\cdot|s_h^t) \right\rangle \mid s_1 \right] \right] \qquad \text{(Linearity of expectation)}$$

$$= \mathbb{E}_c \left[ \sum_{t=1}^{T} \sum_{h=1}^{H} \mathbb{E}_{\pi_c^\star, P_c^\star} \left[ \left\langle \hat{Q}_{c, h}^t(s_h, \cdot), \pi_c^t(\cdot|s_h) - \pi_c^\star(\cdot|s_h) \right\rangle \mid s_1 \right] \right]$$

(The distribution of state $s_h$ in layer $h$ is now independent of $t$, then we use linearity of expectation again)

$$= \mathbb{E}_c \left[ \sum_{h=1}^{H} \mathbb{E}_{\pi_c^\star, P_c^\star} \left[ \sum_{t=1}^{T} \left\langle \hat{Q}_{c, h}^t(s_h, \cdot), \pi_c^t(\cdot|s_h) - \pi_c^\star(\cdot|s_h) \right\rangle \mid s_1 \right] \right]. \qquad (3)$$

Hence, the latter form of term $(iii)$ is the linear term used in Online Mirror Descent (OMD, see Appendix B.1) optimization problem, in expectation over the contexts and states. Also, note that in each iteration of our algorithm, our policy is the solution of the OMD optimization problem with KL divergence, for each $(h, s, c) \in [H] \times S_h \times \mathcal{C}$ separately. Meaning, our policy $\pi_c^{t+1}(\cdot|s)$ satisfies,

$$\pi_c^{t+1}(\cdot|s) \in \arg\min_{\pi \in \Delta_A} \eta \left\langle \hat{Q}_{c,h}^t(s, \cdot), \pi_c(\cdot|s) - \pi_c^t(\cdot|s) \right\rangle + d_{KL}(\pi \| \pi_c^t(\cdot|s)). \tag{4}$$

Thus, similarly to the proof of Lemma 4 in (Shani et al., 2020), we can apply the fundamental inequality of Online Mirror Descent (Theorem B.1) and obtain for any fixed context $c \in \mathcal{C}$, layer $h \in [H]$ and state $s \in S_h$ the following, as our $Q$ estimators are bounded in $[0, H]$.

$$\sum_{t=1}^{T} \left\langle \hat{Q}_{c,h}^t(s, \cdot), \pi_c^t(\cdot|s) - \pi_c(\cdot|s) \right\rangle \leq \frac{\log|A|}{\eta} + \frac{\eta}{2} \sum_{t=1}^{T} \sum_{a \in A} \pi_h^t(a|c, s) \left( \hat{Q}_c^t(s, a) \right)^2$$
$$\leq \frac{\log|A|}{\eta} + \frac{\eta T H^2}{2},$$

which implies that

$$\sum_{h=1}^{H} \mathbb{E}_{\pi_c^\star, P_c^\star} \left[ \sum_{t=1}^{T} \left\langle \hat{Q}_{c,h}^t(s_h, \cdot), \pi_c^t(\cdot|s_h) - \pi_c^\star(\cdot|s_h) \right\rangle \mid s_1 \right] \leq \frac{\log|A|H}{\eta} + \frac{\eta T H^3}{2}.$$

By taking the expectation over the context on both sides we obtain,

$$\mathbb{E}_c \left[ \sum_{h=1}^{H} \mathbb{E}_{\pi_c^\star, P_c^\star} \left[ \sum_{t=1}^{T} \left\langle \hat{Q}_{c,h}^t(s_h, \cdot), \pi_c^t(\cdot|s_h) - \pi_c^\star(\cdot|s_h) \right\rangle \mid s_1 \right] \right] \leq \frac{\log|A|H}{\eta} + \frac{\eta T H^3}{2},$$

which using Equation (3) implies that

$$\sum_{t=1}^{T} \mathbb{E}_{c_t} \left[ \sum_{h=1}^{H} \mathbb{E}_{\pi_{c_t}^\star, P_{c_t}^\star} \left[ \left\langle \hat{Q}_{c_t,h}^t(s_h^t, \cdot), \pi_{c_t}^t(\cdot|s_h^t) - \pi_{c_t}^\star(\cdot|s_h^t) \right\rangle \mid s_1 \right] \right] \leq \frac{\log|A|H}{\eta} + \frac{\eta T H^3}{2}.$$

By choosing $\eta = \sqrt{2\log|A|/H^2 T}$, we obtain that $(iii) \leq \sqrt{2H^4 T \log|A|}$. $\qquad \square$

**Lemma A.3** (Restatement of Lemma 4.4). *With probability at least $1 - \delta/4$,*

$$(iv) \leq H(H+1) + \frac{112 T H^3 \log(128 T^4 H |\mathcal{F}||\mathcal{P}|/\delta^2)}{\beta_\ell} + \frac{25 T H^4 \log(8 T H |\mathcal{P}|/\delta)}{\beta_P}.$$

*Proof.* It holds that

$$(iv) = \sum_{t=1}^{T} \mathbb{E}_{c_t} \left[ \sum_{h=1}^{H} \mathbb{E}_{\pi_{c_t}^\star, P_{c_t}^\star} \left[ \hat{Q}_{c_t,h}^t(s_h^t, a_h^t) - \left( \ell_{c_t}^\star(s_h^t, a_h^t) + P_{c_t}^\star \hat{V}_{c_t,h+1}^t(s_h^t, a_h^t) \right) \mid s_1 \right] \right]$$

$$\leq H(H+1) + \sum_{t=2}^{T} \mathbb{E}_{c_t} \left[ \sum_{h=1}^{H} \mathbb{E}_{\pi_{c_t}^\star, P_{c_t}^\star} \left[ \hat{Q}_{c_t,h}^t(s_h^t, a_h^t) - \left( \ell_{c_t}^\star(s_h^t, a_h^t) + P_{c_t}^\star \hat{V}_{c_t,h+1}^t(s_h^t, a_h^t) \right) \mid s_1 \right] \right]$$
$$\qquad \text{(Since } \hat{Q}_h^t, \hat{V}_h^t \in [0, H] \; \forall h \in [H] \text{ and } \ell^\star \in [0, 1] \text{ the maximal difference is bounded by } H+1)$$

$$= H(H+1) + \mathbb{E}_c \left[ \sum_{t=2}^{T} \sum_{h=1}^{H} \mathbb{E}_{\pi_c^\star, P_c^\star} \left[ \max\left\{ 0, \hat{\ell}_c^t(s_h^t, a_h^t) + \mathbb{E}_{s' \sim \hat{P}_c^t(\cdot|s_h^t, a_h^t)} \left[ \hat{V}_{c,h+1}^t(s') \right] \right\} \right. \right.$$
$$\qquad\qquad\qquad\qquad\qquad\qquad\qquad\qquad\qquad\qquad\qquad \text{(By } \hat{Q}_h^t \text{ definition in Algorithm 1)}$$

$$\left. \left. - \left( \ell_c^\star(s_h^t, a_h^t) + \mathbb{E}_{s' \sim P_c^\star(\cdot|s_h^t, a_h^t)} \left[ \hat{V}_{c,h+1}^t(s') \right] \right) \mid s_1 \right] \right]$$

$$\leq H(H+1) + \mathbb{E}_c\left[\sum_{t=2}^{T}\sum_{h=1}^{H}\mathbb{E}_{\pi_c^\star, P_c^\star}\left[\max\left\{0 - \left(\ell_c^\star(s_h^t, a_h^t) + \mathbb{E}_{s'\sim P_c^\star(\cdot|s_h^t, a_h^t)}\left[\hat{V}_{c,h+1}^t(s')\right]\right),\right.\right.\right.$$

$$\left.\left.\left.\hat{\ell}_c^t(s_h^t, a_h^t) + \mathbb{E}_{s'\sim \hat{P}_c^t(\cdot|s_h^t, a_h^t)}\left[\hat{V}_{c,h+1}^t(s')\right] - \left(\ell_c^\star(s_h^t, a_h^t) + \mathbb{E}_{s'\sim P_c^\star(\cdot|s_h^t, a_h^t)}\left[\hat{V}_{c,h+1}^t(s')\right]\right)\right\}\Big|s_1\right]\right]$$

(By opening the expectation using occupancy measures)

$$= H(H+1) + \mathbb{E}_c\left[\sum_{t=2}^{T}\sum_{h=1}^{H}\sum_{s\in S_h}\sum_{a\in A} q_h(s,a|\pi_c^\star, P_c^\star)\max\left\{\underbrace{0 - \left(\ell_c^\star(s,a) + \mathbb{E}_{s'\sim P_c^\star(\cdot|s,a)}\left[\hat{V}_{c,h+1}^t(s')\right]\right)}_{=:\alpha^t(c,h,s,a)},\right.\right.$$

$$\left.\left.\underbrace{\hat{\ell}_c^t(s,a) + \mathbb{E}_{s'\sim \hat{P}_c^t(\cdot|s,a)}\left[\hat{V}_{c,h+1}^t(s')\right] - \left(\ell_c^\star(s,a) + \mathbb{E}_{s'\sim P_c^\star(\cdot|s,a)}\left[\hat{V}_{c,h+1}^t(s')\right]\right)}_{=:\xi^t(c,h,s,a)}\right\}\right]$$

$$= H(H+1) + \mathbb{E}_c\underbrace{\left[\sum_{t=2}^{T}\sum_{h=1}^{H}\sum_{s\in S_h}\sum_{a\in A} q_h(s,a|\pi_c^\star, P_c^\star)\max\{\alpha^t(c,h,s,a), \xi^t(c,h,s,a)\}\right]}_{=:(\star)}.$$

To bound $(\star)$, we first focus our efforts in bounding $\max\{\alpha^t(c,h,s,a), \xi^t(c,h,s,a)\}$ for any fix context $c \in \mathcal{C}$, round $t \geq 2$, layer $h \in [H]$ and state-action pair $(s,a) \in S_h \times A$.

First, note that $\alpha^t(c,h,s,a) \leq 0$ since $\ell_c^\star(s,a), \hat{V}_{c,h+1}^t(s') \geq 0$.

We thus focus on upper bounding $\xi^t(c,h,s,a)$ with the next-defined $\rho^t(c,h,s,a) \geq 0$. We will use $\rho^t(c,h,s,a)$ to bound the maximum, using the following fact.

**Fact A.4.** *Let $a, b, c \in \mathbb{R}$ such that $a \leq 0$, $c \geq 0$ and $c \geq b$. Then,*

$$\max\{a,b\} \leq \max\{a,b,c\} = \max\{a,c\} = c.$$

Hence, let context $c \in \mathcal{C}$, round $t \geq 2$, layer $h \in [H]$ and state-action pair $(s,a) \in S_h \times A$. The following holds true.

$$\xi^t(c,h,s,a) = \hat{\ell}_c^t(s,a) + \mathbb{E}_{s'\sim \hat{P}_c^t(\cdot|s,a)}\left[\hat{V}_{c,h+1}^t(s')\right] - \left(\ell_c^\star(s,a) + \mathbb{E}_{s'\sim P_c^\star(\cdot|s,a)}\left[\hat{V}_{c,h+1}^t(s')\right]\right)$$

$$= \hat{f}_c^t(s,a) - f_c^\star(s,a) + \left(\hat{P}_c^t(\cdot|s,a) - P_c^\star(\cdot|s,a)\right)\cdot \hat{V}_{c,h+1}^t(\cdot) - b_{\beta_\ell}^t(c,s,a) - 2Hb_{\beta_P}^t(c,s,a)$$

$$\leq |\hat{f}_c^t(s,a) - f_c^\star(s,a)| + H\|\hat{P}_c^t(\cdot|s,a) - P_c^\star(\cdot|s,a)\|_1 - b_{\beta_\ell}^t(c,s,a) - 2Hb_{\beta_P}^t(c,s,a)$$

$$= |\hat{f}_c^t(s,a) - f_c^\star(s,a)| + 2H\frac{1}{2}\|\hat{P}_c^t(\cdot|s,a) - P_c^\star(\cdot|s,a)\|_1 - b_{\beta_\ell}^t(c,s,a) - 2Hb_{\beta_P}^t(c,s,a)$$

(For any two distributions $P, Q$, $TV(P,Q) = \frac{1}{2}\|P-Q\|_1 \leq 1$.)

$$= \min\{1, |\hat{f}_c^t(s,a) - f_c^\star(s,a)|\} + 2H\min\left\{1, \frac{1}{2}\|\hat{P}_c^t(\cdot|s,a) - P_c^\star(\cdot|s,a)\|_1\right\}$$

$$- b_{\beta_\ell}^t(c,s,a) - 2Hb_{\beta_P}^t(c,s,a)$$

$$= \min\left\{1, \sqrt{\frac{\beta_\ell}{\beta_\ell}\frac{1+\sum_{i=1}^{t-1} q_h(s,a|\pi_c^i, \hat{P}_c^t)}{1+\sum_{i=1}^{t-1} q_h(s,a|\pi_c^i, \hat{P}_c^t)}}|\hat{f}_c^t(s,a) - f_c^\star(s,a)|\right\}$$

$$+ 2H\min\left\{1, \sqrt{\frac{\beta_P}{\beta_P}\frac{1+\sum_{i=1}^{t-1} q_h(s,a|\pi_c^i, \hat{P}_c^t)}{1+\sum_{i=1}^{t-1} q_h(s,a|\pi_c^i, \hat{P}_c^t)}}\frac{1}{2}\|\hat{P}_c^t(\cdot|s,a) - P_c^\star(\cdot|s,a)\|_1\right\}$$

$$- b_{\beta_\ell}^t(c,s,a) - 2Hb_{\beta_P}^t(c,s,a)$$

$$\leq \min\left\{1, \frac{\beta_\ell/2}{1+\sum_{i=1}^{t-1} q_h(s,a|\pi_c^i, \hat{P}_c^t)} + \frac{1}{2\beta_\ell} + \frac{1}{2\beta_\ell}\sum_{i=1}^{t-1} q_h(s,a|\pi_c^i, \hat{P}_c^t)\left(\hat{f}_c^t(s,a) - f_c^\star(s,a)\right)^2\right\}$$

(AM-GM + min properties, using that $\left(f^\star - \hat{f}^t\right)^2 \leq 1$, $\|P^\star - \hat{P}^t\|_1^2 \leq 4$.)

$$
\begin{aligned}
&+ 2H \min\left\{1, \frac{\beta_P/2}{1 + \sum_{i=1}^{t-1} q_h(s,a|\pi_c^i, \hat{P}_c^t)} + \frac{1}{2\beta_P} + \frac{1}{2\beta_P} \sum_{i=1}^{t-1} q_h(s,a|\pi_c^i, \hat{P}_c^t)\frac{1}{4}\|\hat{P}_c^t(\cdot|s,a) - P_c^\star(\cdot|s,a)\|_1^2 \right\} \\
&- b_{\beta_\ell}^t(c,s,a) - 2Hb_{\beta_P}^t(c,s,a) \\
\leq &\min\left\{1, \frac{\beta_\ell/2}{1 + \sum_{i=1}^{t-1} q_h(s,a|\pi_c^i, \hat{P}_c^t)}\right\} + \frac{1}{2\beta_\ell} + \frac{1}{2\beta_\ell} \sum_{i=1}^{t-1} q_h(s,a|\pi_c^i, \hat{P}_c^t)\left(\hat{f}_c^t(s,a) - f_c^\star(s,a)\right)^2
\end{aligned}
$$

(Since all terms are positive we can move them outside of the min to increase the sum)

$$
\begin{aligned}
&+ 2H \min\left\{1, \frac{\beta_P/2}{1 + \sum_{i=1}^{t-1} q_h(s,a|\pi_c^i, \hat{P}_c^t)}\right\} + \frac{H}{\beta_P} + \frac{H}{\beta_P} \sum_{i=1}^{t-1} q_h(s,a|\pi_c^i, \hat{P}_c^t)\frac{1}{4}\|\hat{P}_c^t(\cdot|s,a) - P_c^\star(\cdot|s,a)\|_1^2 \\
&- b_{\beta_\ell}^t(c,s,a) - 2Hb_{\beta_P}^t(c,s,a) \\
= &b_{\beta_\ell}^t(c,s,a) + \frac{1}{2\beta_\ell} + \frac{1}{2\beta_\ell} \sum_{i=1}^{t-1} q_h(s,a|\pi_c^i, \hat{P}_c^t)\left(\hat{f}_c^t(s,a) - f_c^\star(s,a)\right)^2 \\
&+ 2Hb_{\beta_P}^t(c,s,a) + \frac{H}{\beta_P} + \frac{H}{\beta_P} \sum_{i=1}^{t-1} q_h(s,a|\pi_c^i, \hat{P}_c^t)\frac{1}{4}\|\hat{P}_c^t(\cdot|s,a) - P_c^\star(\cdot|s,a)\|_1^2 \\
&- b_{\beta_\ell}^t(c,s,a) - 2Hb_{\beta_P}^t(c,s,a) \\
= &\frac{1}{2\beta_\ell} + \frac{1}{2\beta_\ell} \sum_{i=1}^{t-1} q_h(s,a|\pi_c^i, \hat{P}_c^t)\left(\hat{f}_c^t(s,a) - f_c^\star(s,a)\right)^2 \\
&+ \frac{H}{\beta_P} + \frac{H}{\beta_P} \sum_{i=1}^{t-1} q_h(s,a|\pi_c^i, \hat{P}_c^t)\frac{1}{4}\|\hat{P}_c^t(\cdot|s,a) - P_c^\star(\cdot|s,a)\|_1^2.
\end{aligned}
$$

Thus we choose

$$
\begin{aligned}
\rho^t(c,h,s,a) := &\frac{1}{2\beta_\ell} + \frac{1}{2\beta_\ell} \sum_{i=1}^{t-1} q_h(s,a|\pi_c^i, \hat{P}_c^t)\left(\hat{f}_c^t(s,a) - f_c^\star(s,a)\right)^2 \\
&+ \frac{H}{\beta_P} + \frac{H}{\beta_P} \sum_{i=1}^{t-1} q_h(s,a|\pi_c^i, \hat{P}_c^t)\frac{1}{4}\|\hat{P}_c^t(\cdot|s,a) - P_c^\star(\cdot|s,a)\|_1^2,
\end{aligned}
$$

and observe that $\rho^t(c,h,s,a) \geq 0$ and $\rho^t(c,h,s,a) \geq \xi^t(c,h,s,a)$. We recall $\alpha^t(c,h,s,a) \leq 0$ and use Fact A.4 to conclude that

$$
\max\{\alpha^t(c,h,s,a), \xi^t(c,h,s,a)\} \leq \rho^t(c,h,s,a).
$$

We plug the above into $(\star)$ and obtain

$$
\begin{aligned}
(iv) \leq &H(H+1) + \mathbb{E}_c\left[\sum_{t=2}^T \sum_{h=1}^H \sum_{s \in S_h} \sum_{a \in A} q_h(s,a|\pi_c^\star, P_c^\star) \max\{\alpha^t(c,h,s,a), \xi^t(c,h,s,a)\}\right] \\
\leq &H(H+1) + \mathbb{E}_c\left[\sum_{t=2}^T \sum_{h=1}^H \sum_{s \in S_h} \sum_{a \in A} q_h(s,a|\pi_c^\star, P_c^\star)\rho^t(c,h,s,a)\right] \\
= &H(H+1) + \mathbb{E}_c\left[\sum_{t=2}^T \sum_{h,s,a} q_h(s,a|\pi_c^\star, P_c^\star)\left(\frac{1}{2\beta_\ell} \sum_{i=1}^{t-1} q_h(s,a|\pi_c^i, \hat{P}_c^t)\left(\hat{f}_c^t(s,a) - f_c^\star(s,a)\right)^2\right.\right.
\end{aligned}
$$

(We abbreviate by denoting the tree sums as one)

$$
\left.\left. + \frac{1}{2\beta_\ell} + \frac{H}{\beta_P} + \frac{H}{\beta_P} \sum_{i=1}^{t-1} q_h(s,a|\pi_c^i, \hat{P}_c^t)\frac{1}{4}\|\hat{P}_c^t(\cdot|s,a) - P_c^\star(\cdot|s,a)\|_1^2\right)\right]
$$

(Reorder sums using that the occupancy measures in each layer define a distribution and are bounded by 1.)

$$\leq H(H+1) + \frac{TH}{2\beta_\ell} + \frac{TH^2}{\beta_P} + \frac{1}{2\beta_\ell}\mathbb{E}_c\left[\sum_{t=2}^{T}\sum_{i=1}^{t-1}\sum_{h=1}^{H}\sum_{s\in S_h}\sum_{a\in A} q_h(s,a|\pi_c^i,\hat{P}_c^t)\left(\hat{f}_c^t(s,a) - f_c^\star(s,a)\right)^2\right]$$

(Translating occupancy measures to expectation over cumulative loss)

$$+ \frac{H}{\beta_P}\mathbb{E}_c\left[\sum_{t=2}^{T}\sum_{i=1}^{t-1}\sum_{h=1}^{H}\sum_{s\in S_h}\sum_{a\in A} q_h(s,a|\pi_c^i,\hat{P}_c^t)\frac{1}{4}\|\hat{P}_c^t(\cdot|s,a) - P_c^\star(\cdot|s,a)\|_1^2\right]$$

$$= H(H+1) + \frac{TH}{2\beta_\ell} + \frac{TH^2}{\beta_P} + \frac{1}{2\beta_\ell}\mathbb{E}_c\left[\sum_{t=2}^{T}\sum_{i=1}^{t-1}\mathbb{E}_{\pi_c^i,\hat{P}_c^t}\left[\sum_{h=1}^{H}\left(\hat{f}_c^t(s_h^i,a_h^i) - f_c^\star(s_h^i,a_h^i)\right)^2 \mid s_1\right]\right]$$

$$+ \frac{H}{\beta_P}\mathbb{E}_c\left[\sum_{t=2}^{T}\sum_{i=1}^{t-1}\mathbb{E}_{\pi_c^i,\hat{P}_c^t}\left[\sum_{h=1}^{H}\frac{1}{4}\|\hat{P}_c^t(\cdot|s_h^i,a_h^i) - P_c^\star(\cdot|s_h^i,a_h^i)\|_1^2 \mid s_1\right]\right]$$

$$\leq H(H+1) + \frac{TH}{2\beta_\ell} + \frac{TH^2}{\beta_P} + \frac{3}{2\beta_\ell}\mathbb{E}_c\left[\sum_{t=2}^{T}\sum_{i=1}^{t-1}\mathbb{E}_{\pi_c^i,P_c^\star}\left[\sum_{h=1}^{H}\left(\hat{f}_c^t(s_h^i,a_h^i) - f_c^\star(s_h^i,a_h^i)\right)^2 \mid s_1\right]\right]$$

(By the value change of measure lemma (Lemma B.2))

$$+ \frac{9H^2}{2\beta_\ell}\mathbb{E}_c\left[\sum_{t=2}^{T}\sum_{i=1}^{t-1}\mathbb{E}_{\pi_c^i,P_c^\star}\left[\sum_{h=1}^{H}D_H^2\left(\hat{P}_c^t(\cdot|s_h^i,a_h^i), P_c^\star(\cdot|s_h^i,a_h^i)\right) \mid s_1\right]\right]$$

$$+ \frac{3H}{\beta_P}\mathbb{E}_c\left[\sum_{t=2}^{T}\sum_{i=1}^{t-1}\mathbb{E}_{\pi_c^i,P_c^\star}\left[\sum_{h=1}^{H}\frac{1}{4}\|\hat{P}_c^t(\cdot|s_h^i,a_h^i) - P_c^\star(\cdot|s_h^i,a_h^i)\|_1^2 \mid s_1\right]\right]$$

$$+ \frac{9H^3}{\beta_P}\mathbb{E}_c\left[\sum_{t=2}^{T}\sum_{i=1}^{t-1}\mathbb{E}_{\pi_c^i,P_c^\star}\left[\sum_{h=1}^{H}D_H^2\left(\hat{P}_c^t(\cdot|s_h^i,a_h^i), P_c^\star(\cdot|s_h^i,a_h^i)\right) \mid s_1\right]\right]$$

(For any two distributions $Q, P$, $\|P - Q\|_1^2 \leq 4D_H^2(P,Q)$)

$$\leq H(H+1) + \frac{TH}{2\beta_\ell} + \frac{TH^2}{\beta_P} + \frac{3}{2\beta_\ell}\sum_{t=2}^{T}\mathbb{E}_c\left[\sum_{i=1}^{t-1}\mathbb{E}_{\pi_c^i,P_c^\star}\left[\sum_{h=1}^{H}\left(\hat{f}_c^t(s_h^i,a_h^i) - f_c^\star(s_h^i,a_h^i)\right)^2 \mid s_1\right]\right]$$

$$+ \left(\frac{9H^2}{2\beta_\ell} + \frac{9H^3}{\beta_P} + \frac{3H}{\beta_P}\right)\sum_{t=2}^{T}\mathbb{E}_c\left[\sum_{i=1}^{t-1}\mathbb{E}_{\pi_c^i,P_c^\star}\left[\sum_{h=1}^{H}D_H^2\left(\hat{P}_c^t(\cdot|s_h^i,a_h^i), P_c^\star(\cdot|s_h^i,a_h^i)\right) \mid s_1\right]\right]$$

(By union bound over Corollaries 2.1 and 2.3, holds with probability at least $1 - \delta/4$)

$$\leq H(H+1) + \frac{TH}{2\beta_\ell} + \frac{TH^2}{\beta_P} + \frac{3}{2\beta_\ell}68TH\log(16T^3|\mathcal{F}|/\delta) + \left(\frac{9H^2}{2\beta_\ell} + \frac{9H^3}{\beta_P} + \frac{3H}{\beta_P}\right)2TH\log(8TH|\mathcal{P}|/\delta)$$

$$\leq H(H+1) + \frac{112TH^3\log(128T^4H|\mathcal{F}||\mathcal{P}|/\delta^2)}{\beta_\ell} + \frac{25TH^4\log(8TH|\mathcal{P}|/\delta)}{\beta_P},$$

as desired. $\qquad\square$

## A.2. Deriving the Bonus Bound

The most technically significant component of our analysis is the derivation of the bonus bound. To establish this result, we carefully analyze four possible events associated with each fixed context. For three of these event combinations, we derive the desired upper bound on the bonus terms, and then obtain the final bound by taking the expectation over the contexts. The technical details are given below, yielding the proof of Lemma A.11.

Fix any context $c \in \mathcal{C}$. We define two types of events that characterize the quality of the learned dynamics for this context, and then analyze the cumulative bonus associated with $c$ under these events. Let $\gamma, \gamma' > 0$ satisfy $432H^4\gamma \leq \gamma' \leq \frac{1}{2}\min\{\beta_P, \beta_\ell\}$. We consider the following two classes of events.

**Event Type A: *Expected dynamics approximation error.*** For each $t \geq 2$, define $A_c^t$ as the event that the expected dynamics

approximation error measured in squared Hellinger distance at round $t$ exceeds $\gamma$. Formally,

$$A_c^t = \left\{ \sum_{i=1}^{t-1} \sum_{h=1}^{H} \sum_{s \in S_h} \sum_{a \in A} q_h(s,a|\pi_c^i, P_c^\star) D_H^2(\hat{P}_c^t(\cdot|s,a), P_c^\star(\cdot|s,a)) > \gamma \right\}.$$

Its complement event, $\bar{A}_c^t$, indicates that the expected approximation error is at most $\gamma$.

**Event Type B: *Expected visitations.*** For each $t \geq 2$, layer $h \in [H]$, and state-action pair $(s,a) \in S_h \times A$, define $B_c^t(h,s,a)$ as the event that the expected number of visits to $(s,a)$ under context $c$ up to round $t-1$ is at most $\gamma' - 1$. Formally,

$$B_c^t(h,s,a) = \left\{ 1 + \sum_{i=1}^{t-1} q_h(s,a|\pi_c^i, P_c^\star) \leq \gamma' \right\}.$$

The complementary event $\bar{B}_c^t(h,s,a)$ indicates that the expected number of visits exceeds $\gamma' - 1$.

These two types of events jointly quantify how well the dynamics have been approximated relative to the expected number of samples collected for each context-state-action triplet up to time $t$. Intuitively, $A_c^t$ captures cases where the learned dynamics is still inaccurate, while $B_c^t(h,s,a)$ identifies under-explored regions of the context-related MDP. Together, they characterize the amount of exploration still required to refine the dynamics model before reliable exploitation becomes possible, affecting the cumulative cost of the bonuses. We identify that under a complimentary combination of those events, we conclude the useful properties elaborated next. The first property immediately implied by Markov inequality and is as simple as stated below.

**Observation A.5** (Restatement of Observation 4.5). *For any fixed $c \in \mathcal{C}$ and $t \geq 2$ it holds that*

$$\mathbb{I}[A_c^t] \leq \frac{1}{\gamma} \sum_{i=1}^{t-1} \sum_{h=1}^{H} \sum_{s \in S_h} \sum_{a \in A} q_h(s,a|\pi_c^i, P_c^\star) D_H^2(\hat{P}_c^t(\cdot|s,a), P_c^\star(\cdot|s,a)).$$

For the second property, we observe that for any $h \in [H], s \in S_h, a \in A$ that are underexplored at round $t$, meaning $B_c^t(h,s,a)$ holds, the bonus are maximal and have the value of 1. By $\gamma'$ choice, this immediately implies they are also upper bounded using the true occupancy measures, meaning by $\frac{\beta/2}{1+\sum_{i=1}^{t-1} q_h(s,a|\pi_c^i, P_c^\star)}$. This is formally given in the next claim.

**Claim A.6** (Restatement of Claim 4.6). *For any fixed $c \in \mathcal{C}, t \geq 2, h \in [H], s \in S_h, a \in A$ such that event $B_c^t(h,s,a)$ holds (i.e., $\mathbb{I}[B_c^t(h,s,a)] = 1$) the followings hold true.*

$$b_{\beta_\ell}^t(c,s,a) \leq \frac{\beta_\ell/2}{1+\sum_{i=1}^{t-1} q_h(s,a|\pi_c^i, P_c^\star)} \quad \text{and} \quad b_{\beta_P}^t(c,s,a) \leq \frac{\beta_P/2}{1+\sum_{i=1}^{t-1} q_h(s,a|\pi_c^i, P_c^\star)}.$$

*Proof.* Let $c, t, h, s, a$ such that $\mathbb{I}[B_c^t(h,s,a)] = 1$. Observe that for $\beta \in \{\beta_P, \beta_\ell\}$,

1. By the bonuses definition, $b_\beta^t(c,h,s,a) \leq 1$.

2. Since $\gamma' \leq \frac{1}{2} \min\{\beta_P, \beta_\ell\}$, it holds that $\gamma' \leq \beta/2$.

By combining both observations we conclude,

$$\frac{\beta/2}{1+\sum_{i=1}^{t-1} q_h(s,a|\pi_c^i, P_c^\star)} \geq \frac{\beta/2}{\gamma'} \geq \frac{\beta/2}{\beta/2} = 1 \geq b_t^\beta(c,h,s,a),$$

which implies the lemma. $\qquad\square$

**Lemma A.7** (Restatement of Lemma 4.7). *Let any $c \in \mathcal{C}, t \geq 2, h \in [H], s \in S_h, a \in A$ such that both events $\bar{A}_c^t$ and $\bar{B}_c^t(h,s,a)$ hold (i.e., $\mathbb{I}[\bar{A}_c^t] = 1$ and $\mathbb{I}[\bar{B}_c^t(h,s,a)] = 1$). Then, the following holds true.*

$$1 + \sum_{i=1}^{t-1} q_h(s,a|\pi_c^i, \hat{P}_c^t) \geq \frac{1}{6} \left( 1 + \sum_{i=1}^{t-1} q_h(s,a|\pi_c^i, P_c^\star) \right).$$

*Proof.* Let any $(c, t, h, s, a)$ for which both events hold. We define a specific loss function $\ell_{s,a}$ related for the fixed state $s$ and action $a$ as $\ell_{s,a}(s', a') = \mathbb{I}[(s', a') = (s, a)]$. That is, $\ell_{s,a}$ is the loss function that returns 1 if we visit the state $s$ and play the action $a$ in this current time-step $h$ of the episode, and 0 to any other state-action pair. We note that the original CMDP is layered, hence, $(s, a) \in S_h \times A$ are related to a specific layer $h$. We use $\ell_{s,a}$ to define two MDPs related to the tuple $(c, t, h, s, a)$:

(1) $M^t_{s,a}(c) = (S, A, \hat{P}^t_c, \ell_{s,a}, s_1, H)$ which is the MDP with the approximated dynamics at round $t$ and the indicator loss function for state $s$ and action $a$. Then, for all $i \in [t-1]$ it holds that

$$V^{\pi^i_c}_{M^t_{s,a}(c)}(s_1) = \sum_{h'=1}^{H} \sum_{s' \in S_{h'}} \sum_{a' \in A} q_{h'}(s', a' | \pi^i_c, \hat{P}^t_c) \mathbb{I}[(s', a') = (s, a)] = q_h(s, a | \pi^i_c, \hat{P}^t_c).$$

(2) $M^\star_{s,a}(c) = (S, A, P^\star_c, \ell_{s,a}, s_1, H)$ which is the MDP with the true dynamics of the context $c$ and the indicator loss function for state $s$ and action $a$. Similarly, for all $i \in [1, t-1]$ it holds that

$$V^{\pi^i_c}_{M^\star_{s,a}(c)}(s_1) = \sum_{h'=1}^{H} \sum_{s' \in S_{h'}} \sum_{a' \in A} q_{h'}(s', a' | \pi^i_c, P^\star_c) \mathbb{I}[(s', a') = (s, a)] = q_h(s, a | \pi^i_c, P^\star_c).$$

Hence, for all $i \in [t-1]$ we can apply Corollary B.3 of the value change of measure lemma (Lemma B.2) to conclude that

$$q_h(s, a | \pi^i_c, P^\star_c) \le 3 q_h(s, a | \pi^i_c, \hat{P}^t_c) + 216 H^4 \mathop{\mathbb{E}}_{\pi^i_c, P^\star_c} \left[ \sum_{h=1}^{H} D^2_H(P^\star_c(\cdot | s_h, a_h), \hat{P}^t_c(\cdot | s_h, a_h)) \,\middle|\, s_1 \right].$$

Summing over all $i$ and adding $+1$ for both sides of the inequality, we obtain that

$$1 + \sum_{i=1}^{t-1} q_h(s, a | \pi^i_c, P^\star_c) \le 1 + 3 \sum_{i=1}^{t-1} q_h(s, a | \pi^i_c, \hat{P}^t_c) \tag{5}$$
$$+ 216 H^4 \sum_{i=1}^{t-1} \mathop{\mathbb{E}}_{\pi^i_c, P^\star_c} \left[ \sum_{h=1}^{H} D^2_H(P^\star_c(\cdot | s_h, a_h), \hat{P}^t_c(\cdot | s_h, a_h)) \,\middle|\, s_1 \right].$$

Next, since event $\bar{A}^t_c$ holds and $432 H^4 \gamma \le \gamma'$, we translate occupancy measures to expectation to conclude that

$$\sum_{i=1}^{t-1} \mathop{\mathbb{E}}_{\pi^i_c, P^\star_c} \left[ \sum_{h=1}^{H} D^2_H(P^\star_c(\cdot | s_h, a_h), \hat{P}^t_c(\cdot | s_h, a_h)) \,\middle|\, s_1 \right] \le \gamma \le \frac{\gamma'}{432 H^4}.$$

In addition, since event $\bar{B}^t_c(h, s, a)$ holds we have $1 + \sum_{i=1}^{t-1} q_h(s, a | \pi^i_c, P^\star_c) > \gamma'$. We combine the two inequalities above to conclude that

$$216 H^4 \sum_{i=1}^{t-1} \mathop{\mathbb{E}}_{\pi^i_c, P^\star_c} \left[ \sum_{h=1}^{H} D^2_H(P^\star_c(\cdot | s_h, a_h), \hat{P}^t_c(\cdot | s_h, a_h)) \,\middle|\, s_1 \right] \le \frac{1}{2} \left( 1 + \sum_{i=1}^{t-1} q_h(s, a | \pi^i_c, P^\star_c) \right).$$

Plugin the latter in Equation (5) we obtain

$$1 + \sum_{i=1}^{t-1} q_h(s, a | \pi^i_c, P^\star_c) \le 1 + 3 \sum_{i=1}^{t-1} q_h(s, a | \pi^i_c, \hat{P}^t_c) + \frac{1}{2} \left( 1 + \sum_{i=1}^{t-1} q_h(s, a | \pi^i_c, P^\star_c) \right),$$

which implies that

$$\frac{1}{2} \left( 1 + \sum_{i=1}^{t-1} q_h(s, a | \pi^i_c, P^\star_c) \right) \le 1 + 3 \sum_{i=1}^{t-1} q_h(s, a | \pi^i_c, \hat{P}^t_c) \le 3 \left( 1 + \sum_{i=1}^{t-1} q_h(s, a | \pi^i_c, \hat{P}^t_c) \right).$$

By dividing both sides of the inequality by 3 we obain the lemma. $\qquad\square$

To use the properties given in Observation A.5, Claim A.6, and Lemma A.7, we decompose the cumulative bonus sum of the context $c$ into three terms by the complementary combination of the four event classes as given next.

$$\sum_{t=2}^{T}\sum_{h=1}^{H}\sum_{s\in S_h}\sum_{a\in A} q_h(s,a|\pi_c^t, P_c^\star)\left(b_{\beta_\ell}^t(c,s,a) + 2Hb_{\beta_P}^t(c,s,a)\right) \tag{6}$$

$$=\sum_{t=2}^{T}\mathbb{I}[A_c^t]\sum_{h=1}^{H}\sum_{s\in S_h}\sum_{a\in A} q_h(s,a|\pi_c^t, P_c^\star)\left(b_{\beta_\ell}^t(c,s,a) + 2Hb_{\beta_P}^t(c,s,a)\right) \tag{7}$$

$$+\sum_{t=2}^{T}\mathbb{I}[\bar{A}_c^t]\sum_{h=1}^{H}\sum_{s\in S_h}\sum_{a\in A}\mathbb{I}[B_c^t(h,s,a)] q_h(s,a|\pi_c^t, P_c^\star)\left(b_{\beta_\ell}^t(c,s,a) + 2Hb_{\beta_P}^t(c,s,a)\right) \tag{8}$$

$$+\sum_{t=2}^{T}\mathbb{I}[\bar{A}_c^t]\sum_{h=1}^{H}\sum_{s\in S_h}\sum_{a\in A}\mathbb{I}[\bar{B}_c^t(h,s,a)] q_h(s,a|\pi_c^t, P_c^\star)\left(b_{\beta_\ell}^t(c,s,a) + 2Hb_{\beta_P}^t(c,s,a)\right). \tag{9}$$

In the next lemmas sequence, we bound each of the sums separately, using the related property. In the final stage of the proof, we will take an expectation over the context and use dynamics approximation concentration to conclude the final result.

**Lemma A.8** (Restatement of Lemma 4.8). *For any fixed context $c \in \mathcal{C}$,*

$$\sum_{t=2}^{T}\mathbb{I}[A_c^t]\sum_{h=1}^{H}\sum_{s\in S_h}\sum_{a\in A} q_h(s,a|\pi_c^t, P_c^\star)\left(b_{\beta_\ell}^t(c,s,a) + 2Hb_{\beta_P}^t(c,s,a)\right)$$

$$\leq\frac{H+2H^2}{\gamma}\sum_{t=2}^{T}\sum_{i=1}^{t-1}\mathbb{E}_{\pi_c^i, P_c^\star}\left[\sum_{h=1}^{H}D_H^2(\hat{P}_c^t(\cdot|s_h^i, a_h^i), P_c^\star(\cdot|s_h^i, a_h^i)) \mid s_1\right].$$

*Proof.* The followings hold for any fixed context $c \in \mathcal{C}$.

$$\sum_{t=2}^{T}\mathbb{I}[A_c^t]\sum_{h=1}^{H}\sum_{s\in S_h}\sum_{a\in A} q_h(s,a|\pi_c^t, P_c^\star)\left(b_{\beta_\ell}^t(c,s,a) + 2Hb_{\beta_P}^t(c,s,a)\right)$$

$$\leq(1+2H)\sum_{t=2}^{T}\mathbb{I}[A_c^t]\sum_{h=1}^{H}\sum_{s\in S_h}\sum_{a\in A} q_h(s,a|\pi_c^t, P_c^\star) \qquad \text{(The bonuses are bounded in } [0,1])$$

$$=(H+2H^2)\sum_{t=2}^{T}\mathbb{I}[A_c^t] \qquad \text{(The occupancy measures sums to 1 in each layer } h \in [H])$$

$$\leq\frac{H+2H^2}{\gamma}\sum_{t=2}^{T}\sum_{i=1}^{t-1}\sum_{h=1}^{H}\sum_{s\in S_h}\sum_{a\in A} q_h(s,a|\pi_c^i, P_c^\star)D_H^2(\hat{P}_c^t(\cdot|s,a), P_c^\star(\cdot|s,a)) \qquad \text{(By Observation A.5)}$$

$$=\frac{H+2H^2}{\gamma}\sum_{t=2}^{T}\sum_{i=1}^{t-1}\mathbb{E}_{\pi_c^i, P_c^\star}\left[\sum_{h=1}^{H}D_H^2(\hat{P}_c^t(\cdot|s_h^i, a_h^i), P_c^\star(\cdot|s_h^i, a_h^i)) \mid s_1\right],$$

$$\text{(Translating occupancy measures to expectation)}$$

as stated. □

**Lemma A.9** (Restatement of Lemma 4.9). *For any fixed context $c$,*

$$\sum_{t=2}^{T}\mathbb{I}[\bar{A}_c^t]\sum_{h=1}^{H}\sum_{s\in S_h}\sum_{a\in A}\mathbb{I}[B_c^t(h,s,a)] q_h(s,a|\pi_c^t, P_c^\star)\left(b_{\beta_\ell}^t(c,s,a) + 2Hb_{\beta_P}^t(c,s,a)\right)$$

$$\leq|S||A|\log(T+1)(\beta_\ell + 2H\beta_P).$$

*Proof.* The followings hold true for any fixed context $c$.

$$\sum_{t=2}^{T}\mathbb{I}[\bar{A}_c^t]\sum_{h=1}^{H}\sum_{s\in S_h}\sum_{a\in A}\mathbb{I}[B_c^t(h,s,a)] q_h(s,a|\pi_c^t, P_c^\star)\left(b_{\beta_\ell}^t(c,s,a) + 2Hb_{\beta_P}^t(c,s,a)\right)$$

$$\leq \sum_{t=2}^{T} \sum_{h=1}^{H} \sum_{s \in S_h} \sum_{a \in A} \mathbb{I}\big[B_c^t(h,s,a)\big] q_h(s,a|\pi_c^t, P_c^\star)\big(b_{\beta_\ell}^t(c,s,a) + 2H b_{\beta_P}^t(c,s,a)\big)$$

(Since all terms are positive, removing the indicator of event type $\bar{A}_c^t$ increases the sum)

$$\leq \sum_{t=2}^{T} \sum_{h=1}^{H} \sum_{s \in S_h} \sum_{a \in A} q_h(s,a|\pi_c^t, P_c^\star)\mathbb{I}\big[B_c^t(h,s,a)\big] \frac{\beta_\ell/2 + 2H\beta_P/2}{1 + \sum_{i=1}^{t-1} q_h(s,a|\pi_c^i, P_c^\star)} \qquad \text{(By Claim A.6)}$$

$$\leq \left(\frac{\beta_\ell}{2} + \frac{2H\beta_P}{2}\right) \sum_{t=2}^{T} \sum_{h=1}^{H} \sum_{s \in S_h} \sum_{a \in A} \frac{q_h(s,a|\pi_c^t, P_c^\star)}{1 + \sum_{i=1}^{t-1} q_h(s,a|\pi_c^i, P_c^\star)}$$

(Since all terms are positive, removing the indicator of event type $\bar{B}_c^t(h,s,a)$ increases the sum)

$$\leq \left(\frac{\beta_\ell}{2} + \frac{2H\beta_P}{2}\right) \sum_{h=1}^{H} \sum_{s \in S_h} \sum_{a \in A} \sum_{t=1}^{T} \frac{q_h(s,a|\pi_c^t, P_c^\star)}{1 + \sum_{i=1}^{t-1} q_h(s,a|\pi_c^i, P_c^\star)}$$

(Adding the terms related to $t = 1$ (they are all positive) and reorder summing)

$$\leq \left(\frac{\beta_\ell}{2} + \frac{2H\beta_P}{2}\right) \sum_{h=1}^{H} \sum_{s \in S_h} \sum_{a \in A} 2\log(T+1) \qquad \text{(By Lemma B.8)}$$

$$= |S||A|\log(T+1)(\beta_\ell + 2H\beta_P), \qquad \text{(Since the CMDP is layered, } \big|\cup_{h \in [H+1]} S_h\big| = |S|)$$

as desired. $\qquad\qquad\qquad\qquad\qquad\qquad\qquad\qquad\qquad\qquad\qquad\qquad\qquad\qquad\qquad\qquad\qquad\square$

**Lemma A.10** (Restatement of Lemma 4.10). *For any fixed context $c \in \mathcal{C}$,*

$$\sum_{t=2}^{T} \mathbb{I}\big[\bar{A}_c^t\big] \sum_{h=1}^{H} \sum_{s \in S_h} \sum_{a \in A} \mathbb{I}\big[\bar{B}_c^t(h,s,a)\big] q_h(s,a|\pi_c^t, P_c^\star)\big(b_{\beta_\ell}^t(c,s,a) + 2H b_{\beta_P}^t(c,s,a)\big)$$
$$\leq 6(\beta_\ell + 2H\beta_P)|S||A|\log(T+1).$$

*Proof.* For any fixed context $c$ it holds that

$$\sum_{t=2}^{T} \mathbb{I}\big[\bar{A}_c^t\big] \sum_{h=1}^{H} \sum_{s \in S_h} \sum_{a \in A} \mathbb{I}\big[\bar{B}_c^t(h,s,a)\big] q_h(s,a|\pi_c^t, P_c^\star)\big(b_{\beta_\ell}^t(c,s,a) + 2H b_{\beta_P}^t(c,s,a)\big)$$

$$\leq \sum_{t=2}^{T} \sum_{h=1}^{H} \sum_{s \in S_h} \sum_{a \in A} \mathbb{I}\big[\bar{A}_c^t\big]\mathbb{I}\big[\bar{B}_c^t(h,s,a)\big] q_h(s,a|\pi_c^t, P_c^\star) \frac{\beta_\ell/2 + 2H\beta_P/2}{1 + \sum_{i=1}^{t-1} q_h(s,a|\pi_c^i, \hat{P}_c^t)}$$

(By the bouses definition and reorder sums)

$$\leq 6\left(\frac{\beta_\ell}{2} + \frac{2H\beta_P}{2}\right) \sum_{t=2}^{T} \sum_{h=1}^{H} \sum_{s \in S_h} \sum_{a \in A} \mathbb{I}\big[\bar{A}_c^t\big]\mathbb{I}\big[\bar{B}_c^t(h,s,a)\big] \frac{q_h(s,a|\pi_c^t, P_c^\star)}{1 + \sum_{i=1}^{t-1} q_h(s,a|\pi_c^i, P_c^\star)} \qquad \text{(By Lemma A.7)}$$

$$\leq 6\left(\frac{\beta_\ell}{2} + \frac{2H\beta_P}{2}\right) \sum_{t=2}^{T} \sum_{h=1}^{H} \sum_{s \in S_h} \sum_{a \in A} \frac{q_h(s,a|\pi_c^t, P_c^\star)}{1 + \sum_{i=1}^{t-1} q_h(s,a|\pi_c^i, P_c^\star)}$$

(Since all terms are positive, removing the indicators increases the overall sum)

$$\leq 6\left(\frac{\beta_\ell}{2} + \frac{2H\beta_P}{2}\right) \sum_{h=1}^{H} \sum_{s \in S_h} \sum_{a \in A} \sum_{t=1}^{T} \frac{q_h(s,a|\pi_c^t, P_c^\star)}{1 + \sum_{i=1}^{t-1} q_h(s,a|\pi_c^i, P_c^\star)}$$

(Summing from $t = 1$ adds positive terms, and reorder sums)

$$\leq 6(\beta_\ell + 2H\beta_P)|S||A|\log(T+1), \qquad \text{(By Lemma B.8 and } \big|\cup_{h \in [H+1]} S_h\big| = |S|)$$

as stated. $\qquad\qquad\qquad\qquad\qquad\qquad\qquad\qquad\qquad\qquad\qquad\qquad\qquad\qquad\qquad\qquad\qquad\qquad\square$

Finally, to conclude Lemma A.11 we plug the results of Lemmas A.8 to A.10 and take an expectation over the context, as elaborated below.

**Lemma A.11** (Restatement of Lemma 4.2). *With probability at least* $1 - \delta/4$,

$$\mathbb{E}_c\left[\sum_{t=2}^{T}\sum_{h=1}^{H}\sum_{s\in S_h}\sum_{a\in A}q_h(s,a|\pi_c^t,P_c^\star)\big(b_{\beta_\ell}^t(c,s,a)+2Hb_{\beta_P}^t(c,s,a)\big)\right]$$
$$\leq \frac{5184TH^7\log(4TH|\mathcal{P}|/\delta)}{\min\{\beta_P,\beta_\ell\}}+7(\beta_\ell+2H\beta_P)|S||A|\log(T+1).$$

*Proof.* For any fixed context $c$, plug the results of Lemmas A.8 to A.10 into Equation (6) to obtain

$$\sum_{t=2}^{T}\sum_{h=1}^{H}\sum_{s\in S_h}\sum_{a\in A}q_h(s,a|\pi_c^t,P_c^\star)\big(b_{\beta_\ell}^t(c,s,a)+2Hb_{\beta_P}^t(c,s,a)\big)$$
$$\leq \frac{H+2H^2}{\gamma}\sum_{t=2}^{T}\sum_{i=1}^{t-1}\mathbb{E}_{\pi_c^i,P_c^\star}\left[\sum_{h=1}^{H}D_H^2(\hat{P}_c^t(\cdot|s_h^i,a_h^i),P_c^\star(\cdot|s_h^i,a_h^i))\mid s_1\right]$$
$$+7|S||A|\log(T+1)(\beta_\ell+2H\beta_P).$$

By taking an expectation over the context $c$ on both sides of the inequality, we obtain

$$\mathbb{E}_c\left[\sum_{t=2}^{T}\sum_{h=1}^{H}\sum_{s\in S_h}\sum_{a\in A}q_h(s,a|\pi_c^t,P_c^\star)\big(b_{\beta_\ell}^t(c,s,a)+2Hb_{\beta_P}^t(c,s,a)\big)\right]$$
$$\leq \frac{H+2H^2}{\gamma}\sum_{t=2}^{T}\mathbb{E}_c\left[\sum_{i=1}^{t-1}\mathbb{E}_{\pi_c^i,P_c^\star}\left[\sum_{h=1}^{H}D_H^2(\hat{P}_c^t(\cdot|s_h^i,a_h^i),P_c^\star(\cdot|s_h^i,a_h^i))\mid s_1\right]\right]$$
$$+7(\beta_\ell+2H\beta_P)|S||A|\log(T+1)$$
$$\leq \frac{H+2H^2}{\gamma}\sum_{t=2}^{T}2H\log(4TH|\mathcal{P}|/\delta)+7(\beta_\ell+2H\beta_P)|S||A|\log(T+1)$$

$$\text{(By Corollary 2.3, holds with probability at least } 1-\delta/4)$$

$$\leq \frac{6TH^3\log(4TH|\mathcal{P}|/\delta)}{\gamma}+7(\beta_\ell+2H\beta_P)|S||A|\log(T+1).$$

Finally, we recall the conditions on $\gamma',\gamma$:

$$0 < 432H^4\gamma \leq \gamma' \leq \frac{1}{2}\min\{\beta_P,\beta_\ell\}.$$

We thus set $\gamma' = \frac{\min\{\beta_P,\beta_\ell\}}{2}$ and $\gamma = \frac{\min\{\beta_P,\beta_\ell\}}{864H^4}$, which is a choice that satisfies the above and yields

$$\mathbb{E}_c\left[\sum_{t=2}^{T}\sum_{h=1}^{H}\sum_{s\in S_h}\sum_{a\in A}q_h(s,a|\pi_c^t,P_c^\star)\big(b_{\beta_\ell}^t(c,s,a)+2Hb_{\beta_P}^t(c,s,a)\big)\right]$$
$$\leq \frac{5184TH^7\log(4TH|\mathcal{P}|/\delta)}{\min\{\beta_P,\beta_\ell\}}+7(\beta_\ell+2H\beta_P)|S||A|\log(T+1),$$

which holds with probability at least $1 - \delta/4$, as stated. $\qquad\square$

## A.3. Driving the Final Regret Bound

**Theorem A.12** (Restatement of Theorem 3.1). *For any* $\delta \in (0,1)$ *let* $\beta_\ell = \sqrt{\frac{5184TH^7\log(128T^4H|\mathcal{F}||\mathcal{P}|/\delta^2)}{7|S||A|\log(T+1)}}$,
$\beta_P = \sqrt{\frac{5184TH^6\log(128T^4H|\mathcal{F}||\mathcal{P}|/\delta^2)}{14|S||A|\log(T+1)}}$ *and* $\eta = \sqrt{2\log|A|/H^2T}$.

*Then, with probability at least* $1 - 3\delta/4$ *it holds that*

$$\mathbb{E}\mathcal{R}_T \leq O\left(H^4\sqrt{T|S||A|\log(T+1)\log(T^4H|\mathcal{F}||\mathcal{P}|/\delta^2)}\right)$$

*By Azuma-Hoeffding's inequality (Theorem B.6), the above also implies a high probability regret bound of the form*

$$\mathcal{R}_T \leq O\Big(H^4\sqrt{T|S||A|\log(T+1)\log(T^4H|\mathcal{F}||\mathcal{P}|/\delta^2)} + 2H\sqrt{2T\log(8/\delta)}\Big).$$

*which holds with probability at least $1 - \delta$.*

*Proof.* By combining the results of Lemmas A.1 to A.3 and A.11 using union bound we obtain that with probability at least $1 - 3\delta/4$, for $\eta = \sqrt{2\log|A|/H^2T}$, it holds that

$$
\begin{aligned}
\mathbb{E}[\mathcal{R}_T] \leq & H + H(H+1) + \frac{224TH^3\log(128T^4H|\mathcal{F}||\mathcal{P}|/\delta^2)}{\beta_\ell} + \frac{50TH^4\log(8TH|\mathcal{P}|/\delta)}{\beta_P} \\
& + \frac{5184TH^7\log(4TH|\mathcal{P}|/\delta)}{\min\{\beta_P, \beta_\ell\}} + 7(\beta_\ell + 2H\beta_P)|S||A|\log(T+1) + \sqrt{2H^4T\log|A|} \\
\leq & \frac{5184TH^7\log(128T^4H|\mathcal{F}||\mathcal{P}|/\delta^2)}{\beta_\ell} + 7\beta_\ell|S||A|\log(T+1) \\
& \qquad\qquad\qquad\qquad\qquad \text{(We increase the factors so that the terms are balanced)} \\
& + \frac{5184TH^7\log(128T^4H|\mathcal{F}||\mathcal{P}|/\delta^2)}{\beta_P} + 14\beta_P H|S||A|\log(T+1) \\
& + \frac{5184TH^7\log(128T^4H|\mathcal{F}||\mathcal{P}|/\delta^2)}{\min\{\beta_\ell, \beta_P\}} + \sqrt{2H^4T\log|A|} + 2(H+1)^2.
\end{aligned}
$$

For the choice in

$$
\begin{aligned}
\beta_\ell &= \sqrt{\frac{5184TH^7\log(128T^4H|\mathcal{F}||\mathcal{P}|/\delta^2)}{7|S||A|\log(T+1)}}, \\
\beta_P &= \sqrt{\frac{5184TH^6\log(128T^4H|\mathcal{F}||\mathcal{P}|/\delta^2)}{14|S||A|\log(T+1)}},
\end{aligned}
$$

we obtain,

$$\mathbb{E}[\mathcal{R}_T] \leq O\Big(H^4\sqrt{T|S||A|\log(T+1)\log(T^4H|\mathcal{F}||\mathcal{P}|/\delta^2)}\Big),$$

which, by Corollary B.7 implies that

$$\mathcal{R}_T \leq O\Big(H^4\sqrt{T|S||A|\log(T+1)\log(T^4H|\mathcal{F}||\mathcal{P}|/\delta^2)}\Big).$$

The above holds with probability at least $1 - \delta$, as stated. □

## B. Auxillary Lemmas

### B.1. Online Mirror Descent

The *Online Mirror Descent (OMD)* algorithm (Orabona, 2019) provides a general framework for online convex optimization. At each iteration $t$, the learner updates its decision by solving the following optimization problem:

$$x_{t+1} \in \arg\min_{x \in \Delta(d)} \eta\langle g_t, x - x_t\rangle + B_\psi(x, x_t), \qquad (10)$$

where $B_\psi(\cdot, \cdot)$ denotes the Bregman divergence induced by a strictly convex potential function $\psi$. In our setting, $\psi$ corresponds to the negative entropy, so $B_\psi$ reduces to the *KL-divergence*.

The following result provides the key inequality that characterizes the behavior of OMD over the simplex with the KL divergence, which also underlies well-known algorithms such as `Hedge` and `EXP3`.

**Theorem B.1** (Fundamental inequality of Online Mirror Descent, Lemma 16 in (Shani et al., 2020), originally Theorem 10.4 in v1 of (Orabona, 2019)). *Suppose that $g_{k,i} \geq 0$ for all $k = 1, \ldots, K$ and $i = 1, \ldots, d$. Let $C = \Delta_d$ be the probability simplex and let $\eta > 0$ be the learning rate. Running OMD with the KL divergence, learning rate $\eta$, and uniform initialization $x_1 = (1/d, \ldots, 1/d)$, ensures that for any $u \in \Delta_d$,*

$$\sum_{k=1}^{K} \langle g_k, x_k - u \rangle \leq \frac{\log d}{\eta} + \frac{\eta}{2} \sum_{k=1}^{K} \sum_{i=1}^{d} x_{k,i} g_{k,i}^2.$$

## B.2. Value-Difference Lemmas

In this subsection, we state known value-difference results used in our paper.

Although the next two lemmas are stated for reward-based MDPs, they also apply to our setting. Indeed, our loss is bounded in $[0, 1]$ and induces a value function via the same Bellman-equation (Equation (1)) based induction, with loss in place of reward, so the value-difference bounds remain unchanged.

**Lemma B.2** (Value change of measure, Lemma 4.1 in (Levy et al., 2024)). *Let $r : S \times A \to [0, 1]$ be a bounded expected rewards function. Let $P^\star$ and $\hat{P}$ denote two dynamics and consider the MDPs $M = (S, A, P^\star, r, s_1, H)$ and $\widehat{M} = (S, A, \hat{P}, r, s_1, H)$. Then, for any policy $\pi$ it holds that*

$$V_{\widehat{M}}^{\pi}(s_1) \leq 3 V_M^{\pi}(s_1) + 9 H^2 \underset{P^\star, \pi}{\mathbb{E}} \left[ \sum_{h=1}^{H} D_H^2(\hat{P}(\cdot|s_h, a_h), P^\star(\cdot|s_h, a_h)) \, \Big| \, s_1 \right].$$

Next, we use Lemma B.2 to obtain the following corollary.

**Corollary B.3.** *Let $r : S \times A \to [0, 1]$ be a bounded expected rewards function. Let $P^\star$ and $\hat{P}$ denote two dynamics and consider the MDPs $M = (S, A, P^\star, r, s_1, H)$ and $\widehat{M} = (S, A, \hat{P}, r, s_1, H)$. Then, for any policy $\pi$ it holds that*

$$V_M^{\pi}(s_1) \leq 3 V_{\widehat{M}}^{\pi}(s_1) + (54 H^2 + 162 H^4) \underset{P^\star, \pi}{\mathbb{E}} \left[ \sum_{h=1}^{H} D_H^2(\hat{P}(\cdot|s_h, a_h), P^\star(\cdot|s_h, a_h)) \, \Big| \, s_1 \right]$$

*Proof.* By Lemma B.2,

$$V_M^{\pi}(s_1) \leq 3 V_{\widehat{M}}^{\pi}(s_1) + 9 H^2 \underset{\hat{P}, \pi}{\mathbb{E}} \left[ \sum_{h=1}^{H} D_H^2(\hat{P}(\cdot|s_h, a_h), P^\star(\cdot|s_h, a_h)) \, \Big| \, s_1 \right]$$

$$= 3 V_{\widehat{M}}^{\pi}(s_1) + 18 H^2 \underset{\hat{P}, \pi}{\mathbb{E}} \left[ \sum_{h=1}^{H} \frac{1}{2} D_H^2(\hat{P}(\cdot|s_h, a_h), P^\star(\cdot|s_h, a_h)) \, \Big| \, s_1 \right].$$

By applying Lemma B.2 again on $\mathbb{E}_{\hat{P}, \pi} \left[ \sum_{h=1}^{H} \frac{1}{2} D_H^2(\hat{P}(\cdot|s_h, a_h), P^\star(\cdot|s_h, a_h)) \, \Big| \, s_1 \right]$ as a value function with reward given by scaled squared Hellinger distance, that is bounded in $[0, 1]$. we get

$$\underset{\hat{P}, \pi}{\mathbb{E}} \left[ \sum_{h=1}^{H} \frac{1}{2} D_H^2(\hat{P}(\cdot|s_h, a_h), P^\star(\cdot|s_h, a_h)) \, \Big| \, s_1 \right] \leq (3 + 9 H^2) \underset{P^\star, \pi}{\mathbb{E}} \left[ \sum_{h=1}^{H} D_H^2(\hat{P}(\cdot|s_h, a_h), P^\star(\cdot|s_h, a_h)) \, \Big| \, s_1 \right].$$

Plugging that back into the first inequality, we conclude the lemma. $\square$

The next two value-difference lemmas were originally proved for a more general MDP setting in which both the loss function and the transition dynamics may vary with the horizon. Clearly, they also hold in our setting.

**Lemma B.4** (Extended Value Difference, Lemma 1 in Shani et al., 2020). *Let $\pi, \pi'$ be two policies and $M = (S, A, \{p_h\}_{h=1}^H, \{\ell_h\}_{h=1}^H)$ and $M' = (S, A, \{p'_h\}_{h=1}^H, \{\ell'_h\}_{h=1}^H)$ be two MDPs. Let $\hat{Q}_{M,h}^{\pi}(s, a)$ be an approximation of the Q-function of policy $\pi$ on the MDP $M$ for all $h, s, a$ and let $\hat{V}_{M,h}^{\pi}(s) = \left\langle \hat{Q}_{M,h}^{\pi}(s, \cdot), \pi_h(\cdot \mid s) \right\rangle$. Then,*

$$\hat{V}_{M,1}^{\pi}(s_1) - V_{M',1}^{\pi'}(s_1) = \sum_{h=1}^{H} \mathbb{E}_{\pi', p'} \left[ \langle \hat{Q}_{M,h}^{\pi}(s_h, \cdot), \pi_h(\cdot|s_h) - \pi'_h(\cdot|s_h) \rangle | s_1 \right]$$

$$+ \sum_{h=1}^{H} \mathbb{E}_{\pi',p'} \left[ \hat{Q}_{M,h}^{\pi}(s_h, a_h) - \ell_h'(s_h, a_h) - p_h'(\cdot|s_h, a_h)\hat{V}_{M,h+1}^{\pi}(\cdot)|s_1 \right].$$

**Lemma B.5** (Value Difference, Corollary 1 in Shani et al., 2020). *Let $M$, $M'$ be any $H$-finite horizon MDPs. Then, for any two policies $\pi$, $\pi'$ the following holds*

$$V_{M,1}^{\pi}(s) - V_{M',1}^{\pi'}(s) = \sum_{h=1}^{H} \mathbb{E}_{\pi',P'} \left[ \langle Q_{M,h}^{\pi}(s_h, \cdot), \pi_h(\cdot|s_h) - \pi_h'(\cdot|s_h) \rangle |s_1 \right]$$

$$+ \sum_{h=1}^{H} \mathbb{E}_{\pi',P'} \left[ \ell_h(s_h, a_h) - \ell_h'(s_h, a_h) + (p_h(\cdot|s_h, a_h) - p_h'(\cdot|s_h, a_h))V_{M,h+1}^{\pi}|s_h = s \right].$$

### B.3. Concentration Inequalities

**Theorem B.6** (Azuma-Hoeffding's inequality). *Let $(X_i)_{i=1}^{N}$ be a martingale difference sequence with respect to the filtration $(\mathcal{F}_i)_{i=0}^{N}$ such that $|X_i| \leq B$ almost surely for all $i \in [N]$. Then with probability at least $1 - \delta$,*

$$\left| \sum_{i=1}^{N} X_i \right| \leq B\sqrt{2N \log \frac{2}{\delta}}.$$

**Corollary B.7** (High probability regret). *With probability at least $1 - \delta/4$, it holds that*

$$\mathcal{R}_T \leq \mathbb{E}\mathcal{R}_T + 2H\sqrt{2T \log(8/\delta)}.$$

*Proof.* The proof is an immediate application of Theorem B.6, as the contexts are stochastic and are sampled i.i.d. in each episode, and the policies $\{\pi^t\}_{t=1}^{T}$ are determined entirely by the history. Thus, we define

$$X_t := V_{c_t}^{\pi_{c_t}^t}(s_1) - V_{c_t}^{\pi_{c_t}^{\star}}(s_1) - \mathbb{E}_{c_t} \left[ V_{c_t}^{\pi_{c_t}^t}(s_1) - V_{c_t}^{\pi_{c_t}^{\star}}(s_1) \right]$$

and note that $(X_i)_{i=1}^{T}$ is a martingale difference sequence with respect to the filtration $\mathcal{F}_t = \{\sigma^i\}_{i=1}^{t}$ which is the history up to (including) time $t$, for all $t \in [T]$, and $\mathcal{F}_0 := \emptyset$ is the empty history. This is indeed a martingale difference sequence as $\pi^1$ is set to be the random policy and for all $t \geq 2$, $\pi^t$ is determined completely by $\{\hat{f}_i, \widehat{P}_i, \pi^i\}_{i=1}^{t-1} \cup \{\hat{f}^t, \hat{P}^t\}$ that are all determined by the history $\mathcal{F}_{t-1}$ (assuming the oracle break ties by a deterministic decision rule.). By that, we also have

$$\mathbb{E}[X_t \mid \mathcal{F}_{t-1}]$$
$$= \mathbb{E}_{c_t} \left[ V_{c_t}^{\pi_{c_t}^t}(s_1) - V_{c_t}^{\pi_{c_t}^{\star}}(s_1) - \mathbb{E}_{c_t} \left[ V_{c_t}^{\pi_{c_t}^t}(s_1) - V_{c_t}^{\pi_{c_t}^{\star}}(s_1) \right] \mid \mathcal{F}_{t-1} \right]$$
$$= \mathbb{E}_{c_t} \left[ V_{c_t}^{\pi_{c_t}^t}(s_1) - V_{c_t}^{\pi_{c_t}^{\star}}(s_1) \mid \mathcal{F}_{t-1} \right] - \mathbb{E}_{c_t} \left[ V_{c_t}^{\pi_{c_t}^t}(s_1) - V_{c_t}^{\pi_{c_t}^{\star}}(s_1) \mid \mathcal{F}_{t-1} \right]$$
$$= 0.$$

In addition, $|X_t| \leq 2H$ for all $t \in [T]$.

Hence, for any fixed $T \in \mathbb{N}$, when summing over $t = 1, 2, \ldots, T$ we obtain, using Theorem B.6, that the following holds with probability at least $1 - \delta/4$,

$$\left| \sum_{t=1}^{T} X_t \right| \leq 2H\sqrt{2T \log(8/\delta)}.$$

In addition,

$$\left| \sum_{t=1}^{T} X_t \right| = \left| \sum_{t=1}^{T} V_{c_t}^{\pi_{c_t}^t}(s_1) - V_{c_t}^{\pi_{c_t}^{\star}}(s_1) - \mathbb{E}_{c_t} \left[ V_{c_t}^{\pi_{c_t}^t}(s_1) - V_{c_t}^{\pi_{c_t}^{\star}}(s_1) \right] \right|$$

$$= \left| \sum_{t=1}^{T} V_{c_t}^{\pi_{c_t}^t}(s_1) - V_{c_t}^{\pi_{c_t}^\star}(s_1) - \sum_{t=1}^{T} \mathbb{E}_{c_t} \left[ V_{c_t}^{\pi_{c_t}^t}(s_1) - V_{c_t}^{\pi_{c_t}^\star}(s_1) \right] \right|$$

$$= |\mathcal{R}_T - \mathbb{E}\mathcal{R}_T|.$$

Hence, with probability at least $1 - \delta/4$,

$$\mathcal{R}_T \leq \mathbb{E}\mathcal{R}_T + 2H\sqrt{2T\log(8/\delta)},$$

as stated. $\qquad\square$

### B.4. Algebraic Lemmas

**Lemma B.8** (Logarithmic sums bound, Lemma C.1 in (Levy et al., 2024)). *Let $S_t = \lambda + \sum_{k=1}^{t-1} x_k$. Suppose $x_t \in [0, \lambda]$ for all $t \in [T]$. Then*

$$\sum_{t=1}^{T} \frac{x_t}{S_t} \leq 2\log(T+1).$$

