# OpenReview forum: "Near-Optimal Regret for Policy Optimization in Contextual MDPs with General Offline Function Approximation"
_ICML.cc/2026/Conference — ICML 2026 regular_

### Official Review · Reviewer_mdtB · 2026-03-02

**Soundness:** 3
**Presentation:** 3
**Significance:** 3
**Originality:** 3
**Overall Recommendation:** 4
**Confidence:** 3

**Summary:**

This paper studies policy optimization in contextual MDPs under general offline function approximation. It is a purely theoretical work; its main contribution is proving a near-optimal high probability regret bound for an optimistic policy optimization algorithm that achieves optimal dependence on the size of the state and action spaces.

**Compliance With Llm Reviewing Policy:**

Affirmed.

**Final Justification:**

This paper makes a theoretically strong and near-optimal contribution to contextual MDPs. The rebuttal satisfactorily resolved most of my theoretical concerns, and while I still have some reservations about practical performance and implementability, these do not outweigh the paper’s technical soundness, originality, and significance. Overall, I appreciate the paper’s theoretical contribution and would like to maintain my positive evaluation.

**Key Questions For Authors:**

1. Why is the layered, loop-free structure introduced in the analysis? Is it possible to extend the theoretical results without this assumption?

2. The algorithm’s dependence on the decision horizon H is far from optimal. Are there possible ways to improve this dependence?

3. Could you provide minimal numerical results to validate the practical performance of the proposed algorithm?

4. How the offline regression oracle be implemented in practice, and dose it introduce significant computational cost?

**Limitations:**

yes

**Strengths And Weaknesses:**

**Strengths:**

This is a technically solid paper, and the deviation of new exploration bonuses for offline function approximation is novel. The corresponding regret bound has better dependence on the size of the state and action spaces. The main technical novelty is explained clearly.


**Weaknesses:**

1. Although this is a purely theoretical paper, a small toy example would provide insight into practical performance and computational cost. For instance, the offline oracle may become time-consuming as the problem size increases. This paper argues that the proposed algorithm admits lower computational cost; an explicit per-iteration complexity analysis or some numerical results would make this claim more convincing.

2. The dependence on the decision horizon is far from optimal. More explanation of this issue, along with a discussion of the analytical challenges in improving the dependence on H, would be helpful.

3. Some assumptions, such as a loop-free structure, are introduced without clear explanation, which also makes the result difficult to compare with existing work such as [1]. More discussion of whether the regret bound can be extended to loopy MDPs, and what the main theoretical challenges will be helpful.

4. The lower bound should be included in Table 1 to facilitate comparison.

5. The authors cloud provide more general intuitions about why contextual MDPs are more challenging than tabular MDPs in achieving optimal regret, how their bonus terms are designed, and why these bonuses improve the dependence on the size of the state action spaces.

[1] Qian J, Hu H, Simchi-Levi D. Offline oracle-efficient learning for contextual mdps via layerwise exploration-exploitation tradeoff[J]. Advances in Neural Information Processing Systems, 2024, 37: 133743-133775.

---

> ### Author Rebuttal · Authors · 2026-03-30
>
> We thank the reviewer for the constructive feedback. We hope you find our responses helpful.
>
> **Run-time complexity and empirical validation.** Our result is theoretical, and, consistent with previous works (e.g., Qian et al. 2024), we do not aim to provide empirical evaluation. Nevertheless, we next describe a run-time analysis. Assuming the offline regression oracles are efficient, the per-episode computation cost of episode $t$ is linear in $t$ and polynomial in $|S|,|A|,H$. In episode $t$, the computation consists of: (i) solving the offline regression  (ii) occupancy-measure computations under $\hat P_t$, and Q-function computation (iii) exponential-weights updates. The occupancies and Q-function can be computed efficiently given $\pi$ and $\hat P_t$ via standard dynamic programming with cost of $O(|S|^2|A|H)$. The policy updates are then simply closed from OMD updates, performed only for observed context. Overall, the runtime complexity is $O(T^2 |S|^2|A|H)$, hence efficient.
>
> **Efficient Offline regression oracle.** This assumption is standard and appears throughout much of the CMDP and CMAB literature. It is also realistic, since for many high-dimensional parametric function classes the oracle can be computed efficiently: it amounts to solving an optimization problem with a strictly convex loss, which can be handled by standard optimization methods such as SGD. The clearest example is linear regression, where a closed-form solution is available. Hence, in the linear case, for example, the computational complexity is that of inverting a matrix, thus efficient.
>
> **Dependence on $H$.** The current horizon dependence is indeed suboptimal. The source is the double use of the value-change-of-measure lemma (in Corollary B.3), which introduces extra horizon factors in the current proof. We believe this is a proof artifact rather than a fundamental barrier, and that a refined statement or sharper use of this lemma is the main route to improving the $H$-dependence in future work. Finally,  note that, it is common in the literature that Policy Optimization algorithms suffer an increased dependence in $H$ compared to value based algorithms (e.g., Efroni et al. 2020 and more).
>
> **Loop-free structure and comparison to Qian et al. (2024).** The layered loop-free assumption is mainly a representational normalization for finite-horizon episodic MDPs, adopted for technical convenience, and does not limit the generality of the result at all. Moreover, by applying the trivial reduction described next from an MDP with loops to a non-loop MDP, our result in fact holds for the loopy-CMDP case as well. Note that a general finite-horizon episodic MDP can be unrolled by augmenting the state with the layer index, which increases the state-space size by a factor of $H$, and therefore contributes only an additional $\sqrt{H}$ through our $\sqrt{|S|}$ dependence on regret. Thus, the method trivially extends to loopy finite-horizon CMDPs as well. This also clarifies the comparison to Qian et al. (2024): their formulation uses time-varying transitions on a shared state space, whereas we make the layer explicit in the state representation. We therefore view the difference as primarily representational. The substantive effect on the regret comparison is in the horizon dependence; aside from $H$, the dependencies on $|S|, |A|, T, \log(|F||P|)$ are directly comparable, and our bound improves the dependence on $|S|,|A|$ to be optimal.
>
> **Lower bound.** The best-known lower bound for stochastic CMDPs with offline function approximation and known dynamics is presented by (Levy and Mansour, 2023) and assumes the same layered structure. The bound is $\Omega(\sqrt{TH|S||A|\log|G|})$, where $G$ is the rewards function class. Hence, the remaining unclosed gap is the $H^{3.5}$ factor. We will mention that more explicitly in the revised version.
>
> **Additional intuition.** Contextual MDPs are strictly more challenging than tabular MDPs since each context may induce a different MDP in each episode, and contexts may not repeat. Thus, naïvely learning a different MDP for each context is infeasible, and encoding the context into the state can blow up the state space. Hence, the known approaches for tabular MDPs are inapplicable, and generalization across contexts must be obtained using function approximation. Our bonus is designed to measure how well a state-action pair has been learned for the current context through cumulative occupancy, while evaluating past policies under the current learned model. Since this bonus simulates the expected number of visits to each state-action pair, it balances exploration and exploitation well, similarly to the bonuses in tabular MDPs. Hence, it avoids unnecessary exploration, which is the cause of the high $|S|$ and $|A|$ factors in (Qian et al. 2024).
>
> We hope our response addresses your questions. Otherwise, we will be happy to engage further and clarify any concern you may have during the discussion period.

---

> > ### Author Rebuttal · Reviewer_mdtB · 2026-04-01
> >
> > Thank you for the detailed response.  Most of my concerns regarding the theoretical aspects have been addressed. Although I understand that this is a purely theoretical paper, including some numerical results on the algorithm’s performance would provide valuable insight into its practical behavior. Overall, I appreciate the paper’s theoretical contribution and would like to maintain my positive evaluation.

---

### Official Review · Reviewer_kGPe · 2026-03-04

**Soundness:** 3
**Presentation:** 3
**Significance:** 3
**Originality:** 3
**Overall Recommendation:** 4
**Confidence:** 3

**Summary:**

This paper proposes a new policy optimization algorithm for stochastic contextual Markov decision processes (CMDPs) under general offline function approximation. The method can be viewed as an extension of optimistic policy optimization techniques from the bandit setting to the CMDP setting. The proposed approach improves upon existing state-of-the-art regret bounds in terms of their dependence on the sizes of the state space $S$ and action space $A$. The resulting regret guarantee matches the known lower bounds in all parameters except for the time horizon $H$, where a gap remains.

**Compliance With Llm Reviewing Policy:**

Affirmed.

**Final Justification:**

The rebuttal has addressed most of my concerns. Although the proposed framework did not achieve the optimal complexity bound in all parameters yet, I would still like to maintain my previous slightly positive.

**Key Questions For Authors:**

- **Extension to the more general layered setting (Qian et al., 2024).**
    Can your analysis be extended to the setting considered in Qian et al. (2024), where the MDP is **time-dependent** (i.e., transition kernels may vary across layers/steps) and the dynamics may **include loops** (e.g., via shared state space across layers)?

    - If **yes**, what regret bound would you obtain in that setting, and which parts of the proof need to be modified?

    - If **no**, please clarify the main barriers and challenges.

- **Applicability of Qian et al. (2024) to your (more structured) setting.**
    Conversely, can the algorithm and analysis of Qian et al. (2024) be specialized to the loop-free, time-homogeneous CMDP setting studied here?

    - If so, what regret bound would their approach yield under your assumptions, and how does it compare to your main bound (especially in the dependence on $S$, $A$, and $H$)?

    - If not, what aspect of their framework prevents a direct specialization?

- **Lower bounds and the remaining gap in horizon dependence.**
    What is the best-known **lower bound** for stochastic CMDPs with offline function approximation in your setting (or the closest comparable setting in the literature)?
    It would be helpful to include this lower bound in Table 1 so readers can directly assess the remaining gap in the dependence on the horizon $H$.

**Limitations:**

Yes

**Strengths And Weaknesses:**

### Strengths

- The paper improves upon existing state-of-the-art regret bounds, particularly in their dependence on the sizes of the state space $S$ and action space $A$.

- The regret bound matches the known lower bounds in all parameters except for the time horizon $H$, leaving only a gap in the horizon dependence.

- The presentation is clear and well structured. The background material on MDPs and related concepts is sufficiently detailed and accessible, making the paper relatively self-contained.

- The technical derivations appear rigorous and carefully developed. Although I did not verify every proof in full detail, the arguments seem logically consistent and technically sound.


---

### Weaknesses

- The analysis is restricted to time-independent and loop-free MDPs. In contrast, prior work such as Qian et al. (2024) considers a more general class of layered MDPs with time-dependent transitions and potentially looping dynamics. It is therefore unclear to what extent the improved regret bounds stem from algorithmic innovation versus the additional structural restrictions imposed in this work. Clarification on whether the techniques can be extended beyond the loop-free setting would strengthen the contribution.

---

> ### Author Rebuttal · Authors · 2026-03-30
>
> We thank the reviewer for their thoughtful comments. Below we provide our response to the raised concerns.
>
> **Extension to the setting of Qian et al. (2024).**  The setting of Qian et al. (2024) can be handled naively by a standard reduction to our loop-free layered representation. Concretely, one augments the state with the layer index and works on states of the form $(s,h)$. The transition kernel at layer $h$ is then represented by applying $P_h$ on the state augmented with layer $h$, $(s,h)$, while all transitions to states outside the next layer have probability zero. This yields an equivalent loop-free layered CMDP to which our algorithm and analysis apply directly. Under this reduction, the state space increases by a factor of $H$. Therefore, the regret changes only through the dependence on $|S|$, which in our bound appears as $\sqrt{|S|}$. As a result, this conversion introduces at most an additional $\sqrt{H}$ factor in the regret, while leaving the dependence on $|S| ,|A|, T, \log(|F||P|)$ unchanged. We also note that, it is common in the literature that Policy Optimization algorithms suffer an increased dependence in $H$ compared to value based algorithms (e.g., Efroni et al. 2020 and more).
>
> **Applicability of Qian et al. (2024) to our setting.** As they allow layered CMDPs with time-varying transition kernels and a shared state space across layers, whereas we study the more loop-free, time-homogeneous setting, their algorithm and analysis can be specialized to our setting essentially as is.
> Under our assumptions, this specialization would yield the same regret bound reported for LOLIPOP in Table 1, namely $\widetilde O(\sqrt{H^7 |S|^4 |A|^3 T \log(|F||P|/\delta)})$
> whereas our main result is $\widetilde O(\sqrt{H^8 |S||A|T\log(|F||P|/\delta)}).$
> Thus, even when specialized to our more structured setting, their approach still has substantially worse dependence on $|S|$ and $|A$ than our bound.
>
> **Lower bound and remaining gap:** The best-known lower bound for stochastic CMDPs with offline function approximation and known dynamic presented by (Levy and Mansour, 2023) and assumes the same layered structure. The presented lower bound is $\Omega( \sqrt{TH|S||A| \log|G| })$ where $G$ is the function class used to approximate the contextual rewards. Hence, the remaining unclosed gap is the  $H^{3.5}$ factor. We will mention the lower bound more explicitly in the revisited version.
>
> We hope our response satisfied your questions. Otherwise, we will be happy to engage further and clarify any concern you may have during the discussion period.

---

> > ### Author Rebuttal · Reviewer_kGPe · 2026-04-01
> >
> > Thank you to the authors for their response to my questions. From the response, I understand that, under the setting studied in Qian et al. (2024), the proposed approach does not outperform Qian et al. (2024) with respect to the dependence on
> > $H$. Nevertheless, the approach still exhibits favorable dependence on other parameters. Moreover, although the result does not yet attain the lower bound, the gap appears relatively limited.
> >
> > Overall, I will maintain my current slightly positive score.

---

### Official Review · Reviewer_KPiZ · 2026-03-07

**Soundness:** 3
**Presentation:** 3
**Significance:** 2
**Originality:** 2
**Overall Recommendation:** 3
**Confidence:** 3

**Summary:**

This paper studies policy optimization in stochastic Contextual Markov Decision Processes (CMDPs) with general offline function approximation. The authors propose OPO-CMDP, a policy optimization algorithm with optimistic exploration. They prove a regret bound of $\tilde{O}(H^4\sqrt{T|S||A|})$, improving the dependence on the state and action spaces compared to prior work while avoiding additional structural assumptions.

**Compliance With Llm Reviewing Policy:**

Affirmed.

**Final Justification:**

The rebuttal clarifies the framework and theoretical contributions, and I acknowledge the improvement in the dependence on the state–action space. However, the dependence on the horizon $H$ still remains noticeably suboptimal compared to known lower bounds. In addition, the claimed practical advantages (e.g., computational efficiency and ease of implementation) are not supported by empirical evidence. As these concerns remain unresolved, I maintain my original score.

**Key Questions For Authors:**

1. The current regret bound has a relatively large dependence on the horizon ($H^4$). Could the authors provide some intuition on what drives this dependence in the analysis? In particular, does it mainly stem from the exploration bonus design? If so, do the authors think alternative bonus constructions or variance-reduction techniques could potentially improve the horizon dependence?

2. The paper assumes a loop-free CMDP structure. Could the authors clarify which parts of the algorithm design or theoretical analysis require this assumption? In particular, would the approach still work in more general CMDPs with loops?

3. The paper mentions that the proposed algorithm has a simpler implementation and lower per-iteration computational cost compared with prior approaches. Could the authors discuss the computational complexity more concretely? In particular, it would be helpful to clarify the per-episode cost of computing occupancy measures, performing policy updates, and solving the regression oracles, and how these compare with prior algorithms.

**Limitations:**

The discussion of limitations could be expanded. It would be helpful to explicitly discuss the assumptions of the model and their implications for practical applicability.

**Strengths And Weaknesses:**

### **Strengths**

**Strong theoretical guarantees.** The paper proposes a policy optimization algorithm for stochastic CMDPs with general offline function approximation and proves a regret bound of $\tilde{O}(H^4\sqrt{T|S||A|})$, improving the dependence on the state and action spaces compared to prior work.

**Novel technical analysis.** The paper introduces a non-trivial exploration bonus design based on occupancy measures and develops a careful regret analysis to control the estimation errors.


### **Weaknesses**

**More restrictive problem setting.** The theoretical results rely on a loop-free CMDP assumption, which is more restrictive than prior work such as LOLIPOP (Qian et al., 2024) that allows general MDP dynamics with loops.

**Loose horizon dependence.** The regret bound scales as $H^4$, which remains far from the known lower bound and leaves a substantial gap in the dependence on the horizon.

**Lack of empirical evaluation.** The paper focuses on theoretical analysis and does not include experimental results. The work would benefit from empirical evaluation, even on synthetic environments, to illustrate the practical behavior of the proposed algorithm.

---

> ### Author Rebuttal · Authors · 2026-03-30
>
> We thank the reviewer for their effort in  reviewing our work.
>  Below we provide our response to the raised concerns.
>
> **Dependence on $H$.** The relatively high $H$ factors do not stem from the bonus construction itself (which we believe is optimal), but from possibly loose analysis of the expected bonus sum. Concretely, the additional $H$ factors are steamed by the double use of the value-change-of-measure lemma (in Corollary B.3.) while bounding the bonuses sum. A better dependence on $H$ would follow from a refined version of the value-change-of-measure lemma, or from a sharper use of it within the analysis, that avoids paying this double $H^4$ horizon factor. We therefore view improving the value-change-of-measure step as the main route toward obtaining a tighter horizon dependence in future work. Finally,  note that, it is common in the literature that Policy Optimization algorithms suffer an increased dependence in $H$ compared to value based algorithms (e.g., Efroni et al. 2020 and more).
>
> **The loop-free assumption:** We would like to clarify that the loop-free layered assumption is mainly a representational normalization for finite-horizon episodic MDPs. By augmenting the state with the layer index, any such MDP can be unrolled into an equivalent loop-free layered one, increasing $|S|$ by a factor of $H$, and thus contributing only an additional $\sqrt{H}$ factor through our $\sqrt{|S|}$ dependence.Therefore, the loop-free formulation does not restrict the scope of our result. Our algorithm and analysis are written in this layered representation for convenience, but the same approach applies to general finite-horizon episodic CMDPs with loops as well.
>
> **Run-time complexity:** Assuming the offline regression oracles are efficient, the per-episode computation cost of episode $t$ is linear in $t$ and polynomial in $|S|,|A|,H$. In episode $t$, the computation consists of: (i) solving the offline regression to compute estimators  (ii) occupancy-measure computations under ($\hat P_t$), and Q-function computation (iii) closed-form exponential-weights updates. The occupancies $q_h(s,a\mid \pi,P)$ and Q-function can be computed efficiently given $\pi$ and $P$ by solving linear program with $|S|,|A|,H$ constraints so this step is standard dynamic programming with cost of $O(|S|^2|A|H)$. The policy updates are then simply closed from OMD updates, performed only for observed context. Overall, the runtime complexity is in $O(T^2 |S|^2|A|H)$, hence efficient.  Our comparison to LOLIPOP is therefore algorithmic: LOLIPOP requires a computing policy cover, trusted occupancy measures, and offline density-estimation subproblems during refinement, and then samples from the policy cover to compute the played policy. Hence, much more conceptually complicated and run-time costly compared to our method that only uses standard regression oracles plus direct policy updates. We will clarify this point in the revision.
>
> **Empirical evaluation.** The focus of our work is theoretical: our goal is to establish the first regret guarantee for policy optimization in stochastic CMDPs with general offline function approximation. In this respect, our paper is aligned with prior theoretical works (e.g., Qian et al. 2024, Levy et al. 2024, Levy and Mansour 2023) on CMDPs under general function approximation, that do not present empirical evaluation.
>
> We hope our response satisfied your concerns, and you will positively revisit your evaluation of our work in light of that. We will be happy to engage further during discussions otherwise.

---

> > ### Author Rebuttal · Reviewer_KPiZ · 2026-04-01
> >
> > Thank you for the detailed response.
> >
> > The horizon dependence remains $H^4$, which seems worse than prior work such as LOLIPOP and is still far from the known lower bound. Moreover, the paper claims practical advantages, such as computational efficiency and ease of implementation, but provides no empirical evaluation to substantiate these claims. Based on the above, I maintain my original score after the rebuttal.

---

> > > ### Author Response · Authors · 2026-04-02
> > >
> > > Dear Reviewer KPiZ,
> > >
> > > We would like to emphasize that, under our layered loop-free formulation, it holds that $|S| \gg H$. Indeed, even in the minimal case with one state per layer, we have $|S| = H+2$, since layers $0$ and $H+1$ each contain one default state. Moreover, we improve the dependence on $|S|$ and $|A|$ from $\sqrt{|S|^4|A|^3}$ in LOLIPOP to $\sqrt{|S||A|}$ in our work. As for the horizon dependence, LOLIPOP has $\sqrt{H^7}$ while we have $\sqrt{H^8}$, so the gap is only an additional $\sqrt{H}$ factor. In other words, we reduce the dependence from $|S|^2$ to $|S|^{0.5}$ and from $|A|^{1.5}$ to $|A|^{0.5}$, while paying only an additional $\sqrt{H}$ factor, with $\sqrt{H} \ll \sqrt{|S|}$. Hence, our upper bound is clearly stronger, and matches the known lower bound in all parameters except for $H$.
> > >
> > > Regarding experiments, we believe our paper is consistent with prior CMDP literature, which is primarily theoretical and does not include empirical evaluation. We also note that there is currently no standard benchmark for this setting. For this reason, we feel that requiring experiments for acceptance of a theoretical paper, especially in a setting where there is no established precedent for such evaluation, is not justified.
> > >
> > > We thank the reviewer again and hope the above clarifies the significance and strength of our result.

---

### Official Review · Reviewer_qBwQ · 2026-03-13

**Soundness:** 4
**Presentation:** 4
**Significance:** 3
**Originality:** 3
**Overall Recommendation:** 5
**Confidence:** 4

**Summary:**

The paper proposes OPO-CMDP, a policy optimization algorithm for stochastic contextual MDPs (CMDPs) under general offline function approximation with realizable finite classes for losses and dynamics. The key idea is an optimistic policy optimization update with a new exploration bonus that evaluates the cumulative occupancy of all past policies under the current dynamics estimate for the given context. The authors prove a high-probability regret bound, improving the dependence on $|S|$ and $|A|$ over the current state of the art and avoiding additional structure (e.g., Eluder dimension or minimum reachability), at the cost of a loop-free layered assumption and a suboptimal dependence on H.

**Compliance With Llm Reviewing Policy:**

Affirmed.

**Final Justification:**

After reading the rebuttal, I remain supportive of the paper and keep my score at 5. The authors clarified several of my main concerns in a satisfactory way: in particular, the rebuttal makes the loop-free assumption feel more like a representational normalization than a substantive restriction, explains more carefully which parts of the comparison to LOLIPOP are directly comparable and which are not, and gives a reasonable account of how the finite-class analysis could extend to more general complexity measures. I also appreciate the explanation that the suboptimal dependence on H appears to come mainly from the repeated use of the value-change-of-measure step, which helps localize a concrete technical bottleneck rather than leaving the gap unexplained. My reservations are not entirely gone—the setting is still somewhat narrower than prior work, the H-dependence remains loose, and the paper does not provide empirical evidence for the practical attractiveness of the method—but these limitations do not outweigh what I see as the paper’s main contribution: a technically strong and nontrivial extension of optimistic policy optimization to stochastic CMDPs with general offline function approximation, together with improved $|S|$ and $|A|$ dependence over prior work. Overall, I continue to view this as a meaningful theoretical advance, and I maintain my score of 5.

**Key Questions For Authors:**

1. Loop-free assumption: Which parts of the analysis critically rely on the loop-free layered structure? Could your bonus and occupancy-stability lemmas be extended to allow loops or time-varying per-layer dynamics (as in LOLIPOP)?

2. Infinite function classes: Can your analysis be extended from finite F and P to infinite classes using covering numbers or Rademacher complexities (for squared loss) and suitable information-theoretic bounds (for Hellinger)? What would the regret dependence look like?

3. Comparison fairness: Since LOLIPOP allows loops and your setting is loop-free, can you more carefully delineate which dependencies (especially in S, A, H) are directly comparable, and which improvements stem from the structural restriction?

4. Is the assumption of access to an offline regression oracle standard in this literature? How realistic is this assumption in practice?

5. What is the main reason for the worse dependence on H relative to prior work? Moreover, how might this dependence be reduced in future work?

**Limitations:**

The paper does mention some future directions, such as improving the dependence on the horizon H, removing the need for counterfactual occupancy measure computation, and extending the approach to larger-scale settings. However, these points are presented more as future work than as a clear discussion of the method’s present limitations. In particular, I would encourage the authors to add a short dedicated limitations paragraph that explicitly discusses the practical scope of their assumptions, including the layered loop-free CMDP structure, finite context space, realizability, and access to offline regression oracles. It would also be helpful to comment on how restrictive these assumptions are in applications beyond the theorem’s setting.

**Strengths And Weaknesses:**

Strengths

- Introduces the first policy optimization (PO) algorithm for CMDPs with general offline function approximation, bridging an important gap between practical PO methods and CMDP theory.
- Proposes a new confidence-bonus design that evaluates all past policies’ occupancies under the current dynamics estimate, together with a careful event decomposition (A/B events) to control mismatch between estimated and true occupancies without Eluder/min-reachability assumptions.
- Extends and adapts OPO techniques from tabular/linear settings to general function approximation with offline oracles, which is nontrivial.
- The proof strategy is well motivated, showing how the new bonus can be controlled via change-of-measure and stability of occupancies under small Hellinger error and sufficient visitation.
- Improves the dependence on |S| and |A| relative to LOLIPOP while using standard offline oracles, and removes strong structural assumptions used in prior works.

Weaknesses

- The setting is restricted to layered, loop-free episodic CMDPs and assumes the dynamics class respects the true layered structure; LOLIPOP allows loops and time-varying per-layer dynamics. This limits the generality and weakens the direct comparability of $S, A$, and $H$ dependencies across works.
- The $H$ dependence $H^4$ is far from optimal (claimed suboptimal by $H^3.5$ relative to lower bounds).
- No empirical validation is provided, despite claims of simplicity and practical favorability over LOLIPOP; there is no evidence that the proposed bonus and on-demand per-context OMD are computationally attractive in practice.

---

> ### Author Rebuttal · Authors · 2026-03-30
>
> We thank the reviewer for the careful reading and positive assessment of our work.
> Below we provide our response to the raised concerns.
>
> **The Loop-free assumption:** We would like to clarify that the loop-free layered assumption is mainly a representational normalization for finite-horizon episodic MDPs: by augmenting the state with the layer index, any such MDP can be unrolled into an equivalent loop-free layered one, increasing $|S|$ by an  $H$ factor, and thus contributing an additional $\sqrt{H}$​ factor through our $\sqrt{|S|}$ dependence. Moreover, LOLIPOP’s time-varying dynamics $P=(P_h)_{h=1}^H$​ already impose a horizon-indexed layered structure; their formulation keeps a shared state space across layers, whereas ours makes the layer explicit in the state representation. We agree that this means the exact $H$-dependence is not directly comparable, but we emphasize that the loop-free formulation does not restrict the scope of our result, and our algorithm and analysis do not directly depend on this assumption in any way.
>
> **Comparison to LOLIPOP:** We would like to clarify that the presence or absence of loops affects only the horizon dependence in the regret. Therefore, aside from the dependence on $H$, the dependencies on $|S|,|A|, T$, and $\log(|F||P|)$ are directly comparable between our setting and LOLIPOP’s.In particular, our improvement in the $|S|$ and $|A|$ dependence is not due to the loop-free formulation, but reflects a genuine improvement in these terms. Thus, while the horizon dependence is somewhat not comparable, the comparison in all other parameters is fair, and our bound has better dependence on $|S|$ and $|A|$
> Infinite function classes. Our results can be extended beyond finite classes by replacing the cardinality-based terms with appropriate learning-complexity measures, such as Rademacher complexity, covering-number, or other suitable complexity notions depending on the underlying regression guarantees. The regret guarantee would retain exactly the same form as in our main theorem, with the $\log(|F||P|)$ term replaced by the corresponding complexity term for the chosen function classes. In other words, the dependence on $|S|,|A|,H,T$ would remain the same, while the logarithmic dependence on the finite class sizes would be replaced by the appropriate complexity measure. We chose to state the result for finite classes in order to keep the presentation clean and avoid introducing additional technical complexities, but the extension to infinite classes via standard learning complexity measures is natural.
>
> **Dependence on $H$.** We believe the algorithm itself is optimal; however, the main source of the additional $H$ factors in our analysis is the double use of the value-change-of-measure lemma (in Corollary B.3.) while bounding the bonuses sum. A better dependence on $H$ would follow from a refined version of the value-change-of-measure lemma, or from a sharper use of it within the analysis, that avoids paying this double $H^4$ horizon factor. We therefore view improving the value-change-of-measure step as the main route toward obtaining a tighter horizon dependence in future work. Finally,  note that it is common in the literature that Policy Optimization algorithms suffer an increased dependence in $H$ compared to value based algorithms (e.g., Efroni et al. 2020 and more).
>
> **Empirical validation and computational efficiency.** The focus of our work is theoretical: our goal is to establish the first regret guarantee for policy optimization in stochastic CMDPs with general offline function approximation. In this respect, our paper is aligned with prior theoretical works (e.g., Qian et al. 2024) on CMDPs under general function approximation, that do not present empirical evaluation.
> Nevertheless, from an algorithmic standpoint, our method is computationally efficient under the standard assumption of access to an efficient offline regression oracle. Under this assumption, the runtime of our procedure is polynomial in (T, |S|, |A|,) and (H), and considered run-time efficient.
>
> **Offline regression oracle.** This assumption is indeed standard and appears throughout much of the CMDP and CMAB literature. It is also realistic, since for many parametric function classes the oracle can be computed efficiently: it amounts to solving an optimization problem with a strictly convex loss, which can be handled by standard optimization methods such as SGD. The clearest example is linear regression, where a closed-form solution is available.
>
> We hope our response satisfied your questions, and we will be happy to engage farther during discussion otherwise. We thank the reviewer again for supporting our paper.

---

> > ### Author Rebuttal · Reviewer_qBwQ · 2026-04-03
> >
> > Thank you for the detailed rebuttal. It addressed my questions and clarified the points I had been uncertain about. I will therefore keep my score unchanged.

---

### Decision · Program_Chairs · 2026-04-30

**Decision:**

Accept (regular)

**Comment:**

This paper introduces OPO-CMDP, a novel policy optimization algorithm for stochastic Contextual Markov Decision Processes that utilizes general offline function approximation. By designing an exploration bonus based on evaluating the cumulative occupancy of all past policies under the current dynamics estimate, the method achieves a regret bound of $\tilde{O}(H^{4}\sqrt{T|S||A|log(|\mathcal{F}||\mathcal{P}|)})$. This yields the first regret bound with an optimal dependence on the state and action spaces, directly improving upon prior state-of-the-art without relying on strong structural assumptions. Reviewers hold a positive opinion of the paper. Reviewer KPiZ raised the question about $H^4$ dependence yet $H$ is usually much smaller than $S$. Given these, I would recommend acceptance.